# POET-X: Memory-efficient LLM Training by Scaling Orthogonal Transformation

Zeju Qiu [1 2]   Lixin Liu [1]   Adrian Weller [3]   Han Shi [4]   Weiyang Liu [1]

**spherelab.ai/poetx**

## Abstract

Efficient and stable training of large language models (LLMs) remains a core challenge in modern machine learning systems. To address this challenge, Reparameterized Orthogonal Equivalence Training (POET), a spectrum-preserving framework that optimizes each weight matrix through orthogonal equivalence transformation, has been proposed. Although POET provides strong training stability, its original implementation incurs high memory consumption and computational overhead due to intensive matrix multiplications. To overcome these limitations, we introduce POET-X, a scalable and memory-efficient variant that performs orthogonal equivalence transformations with significantly reduced computational cost. POET-X maintains the generalization and stability benefits of POET while achieving substantial improvements in throughput and memory efficiency. In our experiments, POET-X enables the pretraining of billion-parameter LLMs on a single Nvidia H100 GPU, and in contrast, standard optimizers such as AdamW run out of memory under the same settings.

## 1. Introduction

Recent years have witnessed the remarkable progress of large language models (LLMs). Training these models typically demands an enormous amount of computational resources, and the process often remains unstable. The reParameterized Orthogonal Equivalence Training (POET) algorithm (Qiu et al., 2025a) has recently demonstrated strong stability, owing to its spectrum-preserving property. However, POET suffers from poor memory efficiency and runs significantly slower than Adam (Kingma & Ba, 2014) due to the cost of intensive large-scale matrix multiplications.

To address this, we propose POET-X, a fast, scalable, and memory-efficient training algorithm that significantly enhances POET's GPU memory and runtime efficiency while preserving its spectrum-preserving property. At the core of POET lies the orthogonal equivalence transformation, and our key contribution is to make this transformation scalable. This is achieved through a comprehensive analysis and optimization of the GPU memory usage and runtime cost of every computation involved in POET. By fully exploiting POET's inherent sparse-training nature, POET-X achieves extremely efficient memory utilization comparable to parameter-efficient finetuning methods such as LoRA (Hu et al., 2022), while attaining a runtime comparable to Adam (Kingma & Ba, 2014). These results are particularly significant, as POET-X effectively enables the pretraining of billion-parameter LLMs (*e.g.*, Llama-8B) on a single NVIDIA H100 GPU while achieving consistently better performance than the *de facto* AdamW optimizer.

Our underlying motivation for scaling POET arises from the great potential of *sparse training*. POET optimizes orthogonal matrices that transform neurons, and these matrices are generally constrained to be sparse. Such sparsity leads to strong parameter efficiency. However, in the original implementation of POET, this efficiency was not reflected in GPU memory usage, preventing POET from being practically applicable. Our work aims to bridge the gap between parameter and memory efficiency, thereby unlocking the full potential of POET's sparse training.

Specifically, we introduce the following strategies to improve POET's memory efficiency:

- Drawing inspiration from matrix-free methods (Chen, 2005) used for solving large-scale linear systems, we reformulate POET's original weight-centric computation into an input-centric form. This reformulation turns POET into a sequence of linear maps, eliminating unnecessary memory consumption by avoiding the storage of intermediate activations associated with weight matrices.

- Leveraging the block-sparse structure of the orthogonal matrices in POET, we introduce a parallel batch-wise computation strategy for their matrix multiplications.

- We greatly improve the memory efficiency of the Cayley-

---

[1] The Chinese University of Hong Kong [2] MPI for Intelligent Systems [3] University of Cambridge [4] Huawei Technologies. Correspondence to: Weiyang Liu <wyliu@cse.cuhk.edu.hk>.

*Proceedings of the 43$^{rd}$ International Conference on Machine Learning*, Seoul, South Korea. PMLR 306, 2026. Copyright 2026 by the author(s).

Neumann parameterization (CNP) (Qiu et al., 2025a) for representing orthogonal matrices. This is done by storing only half of all skew-symmetric matrices in CNP.

After carefully benchmarking the runtime of POET, we identify the most time-consuming operations and take the following steps to improve them:

- After revisiting the matrix multiplication in CNP, we find that, with proper matrix rearrangement, the total number of matrix multiplications can be reduced. The same computation reduction can be performed in both forward and backward passes of CNP.

- For the permutation operations in POET-X, we exploit memory-efficient ways to implement them and also effectively get rid of multiple permutations by merging permutations to the weight matrices in advance.

- For all the computations in POET-X, we develop specialized CUDA kernels to ensure that both forward and backward passes can be performed efficiently.

The central contribution of this paper lies in developing effective means to scale up orthogonal equivalence transformations. While this scalability is crucial for enabling the memory efficiency of POET-X, the proposed techniques are actually of independent interest for optimizing orthogonal matrices in large-scale settings. We briefly summarize our contributions as follows:

- We carefully examine both forward and backward computations in the original POET and identify multiple dimensions to improve memory and runtime efficiency.

- Compared to the original POET, the proposed POET-X achieves 3x GPU memory reduction and 8x runtime speedup without sacrificing the original POET's strong training stability. POET-X's strong memory efficiency makes it possible for a single Nvidia H100 GPU to pretrain an LLM up to 13B parameters. Our experiments demonstrate that POET-X consistently offers **better-than-AdamW performance** and **LoRA-level GPU memory efficiency**.

## 2. Preliminaries of POET

POET reparameterizes each neuron as $W_{RP} = RW_0P$, where $W_0 \in \mathbb{R}^{m \times n}$ is a fixed random weight matrix, and $R \in \mathbb{R}^{m \times m}$, $P \in \mathbb{R}^{n \times n}$ are trainable orthogonal matrices. This formulation performs an orthogonal equivalence transformation (OET) on $W_0$, defined as $\text{OET}(W; R, P) = RWP$ which multiplies $W$ by orthogonal matrices from both sides. The forward pass of POET is thus

$$y = W_{RP}^\top x = (RW_0P)^\top x,$$
$$\text{s.t. } \{R^\top R = RR^\top = I, \ P^\top P = PP^\top = I\}. \quad (1)$$

After training, $R$ and $P$ can be merged into $W_{RP}$, ensuring that POET-trained networks have the no inference overhead.

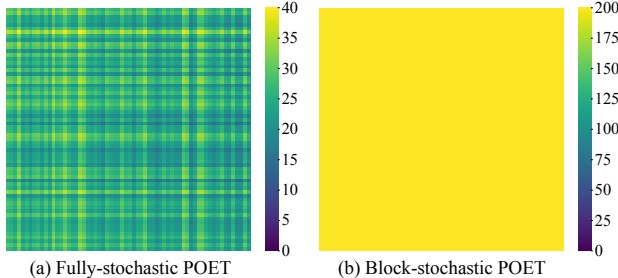

(a) Fully-stochastic POET    (b) Block-stochastic POET

Figure 1. Fully-stochastic POET (with $b = 1/8$) vs. block-stochastic POET (with $b = 8$) for the weight matrix update coverage. In the toy experiment, we use a $64 \times 64$ weight matrix and run both POET variants for a 100-step update, so 200 are the largest possible update steps (multiplication by $R$ and $P$ counts as two updates). Block-stochastic POET ensures balanced update for the weight matrix while fully-stocahstic POET does not.

**Spectrum preservation**. POET can be viewed as learning weight matrices by jointly transforming their left and right singular vectors while keeping the singular values fixed. Given the singular value decomposition (SVD) $W_0 = U\Sigma_0 V^\top$, the reparameterized weight matrix is expressed as $W_{RP} = RU\Sigma_0 V^\top P$, where both $RU$ and $V^\top P$ are orthogonal. This construction effectively forms another SVD of $W_{RP}$, ensuring that its spectral properties remain identical to those of the original matrix $W_0$. More interestingly, if we use zero-mean isotropic Gaussian to initialize $W_0$, then the hyperspherical energy (Liu et al., 2018; 2021b) is provably small due to its invariance under orthogonal transformation (Liu et al., 2021a;b; Qiu et al., 2025a). The properties of spectrum preservation and provably small hyperspherical energy guarantee POET's training stability.

## 3. POET-X: Fast, Memory-efficient Training by Scaling Orthogonal Transformation

We build POET-X on top of block-stochastic POET due to its uniform coverage of all dimensions of the weight matrix. This property is important for memory efficiency, because it ensures the balanced weight update even with a very small number of parameters, unlike the fully-stochastic POET (see Figure 1). In block-stochastic POET, for the $i$-th iteration, the orthogonal matrix $R_i$ is parameterized by

$$R_i = \underbrace{\Psi_i^\top}_{\text{Column-permute}} \cdot \underbrace{\text{Diag}(\tilde{G}_i^1, \tilde{G}_i^2, \cdots, \tilde{G}_i^{\lceil \frac{m}{b} \rceil})}_{\text{Orthogonal matrix } G_i} \cdot \underbrace{\Psi_i}_{\text{Row-permute}} \quad (2)$$

in which $\tilde{G}_i^j \in \mathbb{R}^{b \times b}$ is the $j$-th block of the block-diagonal orthogonal matrix $G_i$, and $\Psi_i, \forall i$ are all random permutation matrices. $P_i$ is also parameterized the same way. Similarly to POET, POET-X optimizes weight matrices by multiplying $R_i$ and $P_i$ into $W_{i-1}$ after every certain number of iterations (i.e., performing the weight update $W_i = R_i W_{i-1} P_i$). Therefore, how to perform these multiplications with the orthogonality constraint on $R_i, P_i$ in a fast and memory-efficient way is our central challenge.

## 3.1. Input-centric Implementation

The original implementation of POET directly operates on the weight matrix $W$ (*i.e.*, $W \leftarrow R_i W P_i$), which is a weight-centric formulation. Despite simplicity, the weight-centric implementation incurs $\mathcal{O}(nm^2)$ complexity (assuming an input vector $x \in \mathbb{R}^m$). Moreover, when computing the gradient *w.r.t.* $R_i$ and $P_i$, both gradients require accessing the weight matrix $W$, thus increasing the memory.

Inspired by matrix-free computation in solving large-scale linear systems, we use an input-centric formulation to implement the update $W \leftarrow R_i W P_i$, as shown below

$$
\underbrace{\underbrace{P_i^\top W^\top}_{\text{① matrix-matrix mult.}} R_i^\top x}_{\text{③ matrix-vector mult.}} \Leftrightarrow \underbrace{P_i^\top \overbrace{W^\top}^{\text{② matrix-vector mult.}} \underbrace{R_i^\top x}_{\text{① matrix-vector mult.}}}_{\text{③ matrix-vector mult.}} \quad (3)
$$

where the left is the weight-centric formulation, which requires two matrix-matrix multiplications and one matrix-vector multiplication, and the right is the input-centric formulation which requires three matrix-vector multiplications. The input-centric formulation has also been shown effective in orthogonal finetuning (Qiu et al., 2025b). However, unlike orthogonal finetuning, which only has one orthogonal matrix $R_i$ to learn (without the need to access $W$), POET-X has an additional orthogonal matrix $P_i$ on the left of $W$ in the above formula. It is highly nontrivial to achieve memory efficiency in this case, because computing the gradient *w.r.t.* $P_i$ still requires accessing $W$, incurring large memory consumption and computational overhead.

## 3.2. Permutation Acceleration and Reduction

To address this challenge, we start by writing out the complete inference formula for one weight matrix:

$$
z = \Phi_n G_P^\top \Phi_n^\top W \Phi_m G_R^\top \Phi_m^\top x \quad (4)
$$

where $R = \Phi_m^\top G_R \Phi_m$ and $P = \Phi_n^\top G_P \Phi_n$. We simplify the notation by dropping the iteration index $i$. Because the inference involves the multiplication of four permutation matrices, we focus on reducing their memory cost.

**Permutation acceleration**. To avoid explicitly constructing the permutation matrices, we implement our customized CUDA operator to perform the permutation. The key idea is to implement an index mapping. Let $W \in \mathbb{R}^{m \times n}$ be the original matrix, $W_{i,:}$ be the $i$-th row of $W$, and $W_{:,j}$ denote the $j$-th column. We define two base index sets $\mathcal{I}_m = \{1, 2, \cdots, m\}$ and $\mathcal{I}_n = \{1, 2, \cdots, n\}$. Then a permutation of indices $\pi$ is defined as $\pi(\mathcal{I}_m) = \{\pi(1), \pi(2), \cdots, \pi(m)\}$ where $\pi : \mathcal{I} \to \mathcal{I}$ is a bijection. $\pi^{-1}$ defines the inverse mapping of $\pi$, and therefore we have $\pi(\pi^{-1}(i)) = i$. Let $\Psi_m \in \{0, 1\}^{m \times m}$ be the permutation matrix defined by the permutation $\pi_p$, and $\Psi_n \in \{0, 1\}^{n \times n}$

| Hidden Dim. | PyTorch (ms) | Ours (ms) | Speedup |
|---|---|---|---|
| 2048 | 1.621 | 0.086 | 18.75× |
| 4096 | 3.269 | 0.167 | 19.57× |
| 8192 | 6.584 | 0.393 | 16.76× |
| 16384 | 13.246 | 0.946 | 14.00× |

Table 1. Comparison between PyTorch-native and our customized permutation under different hidden dimensions.

| Hidden Dim. | 4 permute (ms) | 2 permute (ms) | Speedup |
|---|---|---|---|
| *Block size = 256, Compiled = False* | | | |
| 4096 | 15.858 | 12.017 | 1.32× |
| 8192 | 38.583 | 31.028 | 1.24× |
| 16384 | 120.869 | 94.280 | 1.28× |
| *Block size = 256, Compiled = True* | | | |
| 4096 | 8.705 | 7.496 | 1.16× |
| 8192 | 26.272 | 22.980 | 1.14× |
| 16384 | 90.802 | 78.863 | 1.15× |
| *Block size = 512, Compiled = False* | | | |
| 4096 | 23.592 | 13.421 | 1.76× |
| 8192 | 42.062 | 34.398 | 1.22× |
| 16384 | 126.400 | 110.007 | 1.15× |
| *Block size = 512, Compiled = True* | | | |
| 4096 | 10.235 | 8.986 | 1.14× |
| 8192 | 29.321 | 26.012 | 1.13× |
| 16384 | 96.798 | 86.380 | 1.12× |

Table 2. Runtime improvement of permutation reduction.

be the permutation matrix defined by the permutation $\pi_q$. Then we have the following equations that always hold true:

$$
\begin{aligned}
\Psi_m W &\equiv W' &\Leftrightarrow (W')_{i,:} &= W_{\pi_p(i),:} \\
\Psi_m^T W &\equiv W' &\Leftrightarrow (W')_{i,:} &= W_{\pi_p^{-1}(i),:} \\
W \Psi_n &\equiv W' &\Leftrightarrow (W)_{:,j} &= W_{:,\pi_q^{-1}(j)} \\
W \Psi_n^T &\equiv W' &\Leftrightarrow (W')_{:,j} &= W_{:,\pi_q(j)}.
\end{aligned} \quad (5)
$$

Therefore, to perform the multiplication between the permutation matrix and the weight matrix, we only need to store this permutation index set and access the weight matrix in a prescribed order. Such bijection mapping can be directly used for both forward and backward computation. We conduct an experiment in Table 1 (setup given in Appendix A) to show the acceleration performance of our customized CUDA operator. The results show that the customized CUDA operator is effective with up to 20× speedup.

**Permutation reduction**. In the input-centric formulation of POET, the forward pass requires 4 permutations in total: $\pi_p$, $\pi_q$, $\pi_p^{-1}$ and $\pi_q^{-1}$. We find that 2 permutations can be merged to the weight matrix $W$ in advance:

$$
z = \Phi_n G_P^\top \underbrace{\Phi_n^\top W \Phi_m}_{\text{Pre-computed by permuting } W} G_R^\top \Phi_m^\top x. \quad (6)
$$

In the inner loop of optimizing orthogonal matrices $G_P$ and $G_R$, we can pre-compute the permuted $W$ at the beginning since $W$ stays fixed in the inner loop. The pre-computation can avoid repeated permutation when learning $G_P$ and $G_R$. We empirically compare the runtime of performing the full 4 permutations all the time and the proposed permutation

| Seq. Length | PyTorch (ms) | Ours (ms) | Speedup |
|---|---|---|---|
| 2048 | 0.790 | 0.334 | 2.37× |
| 4096 | 1.525 | 0.642 | 2.38× |
| 8192 | 2.918 | 1.266 | 2.30× |
| 16384 | 6.120 | 2.661 | 2.30× |

Table 3. Runtime comparison between PyTorch-native block-diagonal matrix construction/multiplication and our batch-wise matrix multiplication strategy under different sequence lengths.

| Seq. Length | PyTorch (MB) | Ours (MB) | Reduction |
|---|---|---|---|
| 2048 | 280 | 192 | 31.43% |
| 4096 | 360 | 272 | 24.44% |
| 8192 | 520 | 432 | 16.92% |
| 16384 | 840 | 752 | 9.55% |

Table 4. GPU memory comparison between PyTorch-native block-diagonal matrix construction/multiplication and our batch-wise matrix multiplication strategy under different sequence lengths.

reduction strategy in Table 2 (setup given in Appendix A). The results well validate its effectiveness.

## 3.3. Batch Parallel Computation for Block-diagonal Matrix Multiplication

In the original implementation of block-stochastic POET, all orthogonal matrices adopt a block-diagonal sparse structure:

$$\boldsymbol{G}_P = \mathrm{Diag}(\tilde{\boldsymbol{G}}_P^1, \cdots, \tilde{\boldsymbol{G}}_P^{\lceil \frac{n}{b} \rceil}), \quad \boldsymbol{G}_R = \mathrm{Diag}(\tilde{\boldsymbol{G}}_R^1, \cdots, \tilde{\boldsymbol{G}}_R^{\lceil \frac{m}{b} \rceil}). \quad (7)$$

Consequently, the algorithm must construct numerous large yet sparse orthogonal matrices before performing matrix multiplications. However, we observe that, for block-diagonal matrices, multiplications occur only within each block, making it unnecessary to construct the full block-diagonal matrices in the first place. Motivated by this observation, we propose a batch-parallel strategy, in which we skip the explicit construction of block-diagonal matrices and instead treat each block as an independent matrix, performing batch-wise matrix multiplications accordingly. As shown in Table 3 and Table 4, our batch-parallel strategy not only saves GPU memory but also improves runtime.

## 3.4. Efficient Cayley-Neumann Parameterization

Efficiently ensuring that each diagonal block (*e.g.*, $\tilde{\boldsymbol{G}}_P^i, \forall i$, $\tilde{\boldsymbol{G}}_R^j, \forall j$) in Eq. 7 remains orthogonal during training poses a major challenge. The original POET addresses this by introducing the Cayley-Neumann Parameterization (CNP), which approximates the matrix inverse in the Cayley transform using a Neumann series. CNP improves numerical efficiency at the cost of a slight loss of orthogonality; however, empirical results show that this minor deviation does not affect performance (Qiu et al., 2025a).

The orthogonalization procedure in POET consists of two steps: `skew_symmetric` and `cayley_neumann`. In `skew_symmetric`, it will construct a skew-symmetric matrix $\boldsymbol{Q} \in \mathbb{R}^{b \times b}$ which satisfies $\boldsymbol{Q} = -\boldsymbol{Q}^\top$. Then in

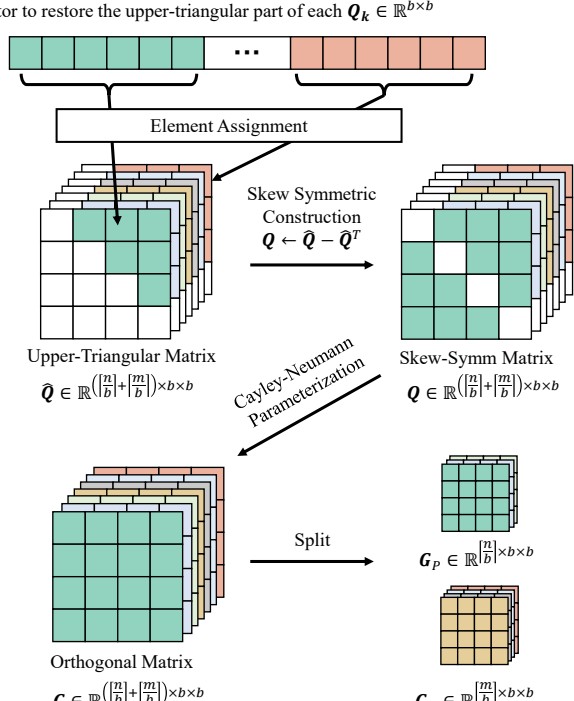

Figure 2. Illustration of efficient Cayley-Neumann parameterization (batch-wise implementation).

`cayley_neumann`, it will turn a skew-symmetric matrix to an (approximately) orthogonal matrix. Based on empirical results in Qiu et al. (2025a), $k = 3$ achieves a good accuracy-efficiency trade-off. Therefore, we consider the CNP with $k = 3$ to construct an orthogonal matrix $\boldsymbol{G}$:

$$\boldsymbol{G} \approx (\boldsymbol{I} + \boldsymbol{Q})(\boldsymbol{I} + \sum_{i=1}^{3} \boldsymbol{Q}^i) = \boldsymbol{I} + 2\boldsymbol{Q} + 2\boldsymbol{Q}^2 + 2\boldsymbol{Q}^3 + \boldsymbol{Q}^4, \quad (8)$$

which is used to replace orthogonal matrices during training and effectively converts a constrained optimization problem into an unconstrained one.

In POET-X, we propose to store skew-symmetric matrices only with their upper-triangular part. The parameter count of $\boldsymbol{Q}$ is $b(b-1)/2$ compared to the original $b^2$. Consequently, the optimizer states and gradients are computed based on this compact representation rather than the full matrix, effectively reducing the POET-related memory footprint by half. This corresponds to the new `skew_symmetric` operation in POET-X, which efficiently constructs the skew-symmetric matrix $\boldsymbol{Q}$ from the actual trainable parameters. Then we consider the `cayley_neumann` operation. We re-arrange the terms in CNP ($k = 3$) to the following form:

$$\boldsymbol{G} \approx 2(\boldsymbol{Q} + \boldsymbol{Q}^2 + \boldsymbol{Q}^2 \cdot \boldsymbol{Q}) + \boldsymbol{Q}^2 \cdot \boldsymbol{Q}^2 + \boldsymbol{I}, \quad (9)$$

from which we observe that all downstream computations depend solely on $\boldsymbol{Q}$ and $\boldsymbol{Q}^2$. This observation reveals a key opportunity for better efficiency. We leverage this dependency through kernel fusion, a simple yet powerful strategy:

| N | PyTorch (ms) | Triton (ms) | Speedup |
|---|---|---|---|
| *Block Size = 256* | | | |
| 64 | 0.316 | 0.107 | 2.96× |
| 128 | 0.610 | 0.204 | 2.99× |
| 192 | 0.881 | 0.297 | 2.97× |
| 256 | 1.156 | 0.388 | 2.98× |
| 320 | 1.404 | 0.479 | 2.93× |
| *Block Size = 512* | | | |
| 64 | 1.308 | 0.660 | 1.98× |
| 128 | 2.477 | 1.296 | 1.91× |
| 192 | 3.659 | 1.937 | 1.89× |
| 256 | 4.825 | 2.569 | 1.88× |
| 320 | 6.069 | 3.249 | 1.87× |

Table 5. Runtime comparison between PyTorch-native implementation of the Cayley-Neumann parameterization and our optimized Triton kernel under different a number of blocks in POET-X.

the two tensors ($Q$ and $Q^2$) are loaded only once into the GPU's low-latency shared memory. From this local cache, we compute higher-order terms ($Q^3$, $Q^4$) and perform the final summation within a single Triton (Tillet et al., 2019) kernel. Compared to a naive PyTorch implementation that repeatedly reads $Q$ and $Q^2$ from the slower global GPU memory for each computation in $G$, our approach drastically reduces data transfer overhead. Besides, the fusion of multiple tensor operations into one custom kernel reduces the number of PyTorch operator calls, consequently improving the kernel launching time on the CPU.

Beside the forward pass, we find that the same strategy can also be applied in the backward pass. Given a function $f(G(Q))$, the gradient *w.r.t.* $Q$ is given by:

$$\nabla_1 = \frac{\partial f}{\partial G}, \quad \nabla_2 = \nabla_1 Q^\top + Q^\top \nabla_1,$$
$$\nabla_3 = \nabla_1 (Q^2)^\top + Q^\top \nabla_2, \quad \nabla_4 = \nabla_2 (Q^2)^\top + (Q^2)^\top \nabla_2,$$
$$\frac{\partial f}{\partial Q} = 2\nabla_1 + 2\nabla_2 + 2\nabla_3 + \nabla_4$$
$$= 2(\nabla_1 + \nabla_2) + (2Q^\top + (Q^2)^\top)\nabla_2 + (2\nabla_1 + \nabla_2)(Q^2)^\top$$

from which we observe that the gradient of $f$ *w.r.t.* $Q$ depends on $Q$, $Q^2$, as well as the gradient tensors $\nabla_1$ and $\nabla_2$. This dependency enables a similar strategy for shared-memory reuse and computation fusion. In our implementation, both the forward and backward kernels (including the batch-based optimizations) are implemented as customized Triton operators. This design enables fine-grained control over GPU memory access and computation, resulting in a substantial runtime speedup. To evaluate the new efficient CNP, we empirically compare our Triton implementation with the PyTorch-native implementation in Table 5. The results show 2-3× speed up, validating its effectiveness.

**Batch-wise CNP implementation**. In practice, we implement both `skew_symmetric` and `cayley_neumann` operations in a batch-wise manner. This can further improve the runtime. An illustration is given in Figure 2.

## 3.5. Boosting Memory-efficiency with Checkpointing

The efficiency of POET-X is substantially enhanced by its input-centric implementation, which incorporates various optimizations and custom kernels. To simplify the subsequent analysis, we omit the permutation matrices from Equation 4. This simplification is justified because our custom permutation kernel incurs no additional memory overhead. This leads to the following simplified forward pass: $z = G_P^\top W G_R^\top x$. We know that for POET training, the central weight matrix $W$ does not require gradients, while the structured matrices $G_P$ and $G_R$ are the ones to be optimized. The input is $x$ and the output is $z$. The forward pass is executed as a sequence of three matrix multiplications:

$$\textbf{mm1}: a = G_R^\top x, \quad \textbf{mm2}: b = W a, \quad \textbf{mm3}: z = G_P^\top b. \quad (10)$$

PyTorch Autograd Engine requires saving intermediate activations during the forward pass to enable the backward pass. These saved tensors are one of the major contributors to the peak memory consumption. We thus begin by analyzing which additional activations have to be saved:

- **mm3 backward** computes $\nabla_{G_P} = b\nabla_z^\top$ and $\nabla_b = G_P \nabla_z$. To compute the gradient for the parameters (*i.e.*, $\nabla_{G_P}$), the Autograd engine must have saved the activation $b$ from the forward pass. This will result in saving an additional tensor of shape $\mathbb{R}^{N \times m}$.

- **mm2 backward** computes $\nabla_a = W^\top \nabla_b$. Since $W$ has no gradient, the activation $a$ is not required to compute the gradient for $W$. Therefore, there is no additional activation needed to be saved.

- **mm1 backward** computes $\nabla_{G_R} = x\nabla_a^\top$ and $\nabla_x = G_R \nabla_a$. To compute $\nabla_{G_R}$, the Autograd engine needs $x$, which is the original input and already available. For $\nabla_x$, the tensor $G_R$ is already in the memory.

We introduce two variants for POET-X that has different memory efficiency. The first, POET-X$_{\text{fast}}$, follows standard Autograd logic, which necessitates saving an additional activation tensor during the forward pass. The second variant, POET-X$_{\text{mem}}$, employs gradient checkpointing to circumvent this memory cost by recomputing the required tensor on-the-fly during the backward pass, making it our most memory-efficient version. POET-X$_{\text{fast}}$ is faster and can be used when the memory is not a critical limitation. We provide an extensive ablation study to the compute-memory trade-off between these two variants in Section 4.

## 3.6. POET-XQ: Quantized POET-X Training

With custom CUDA kernels for both POET-X forward and backward passes, POET-X can readily support quantized training. The core idea is to store only the base model's low-bit quantized weight matrices and dequantize them on the fly whenever needed, such that activation involving high-

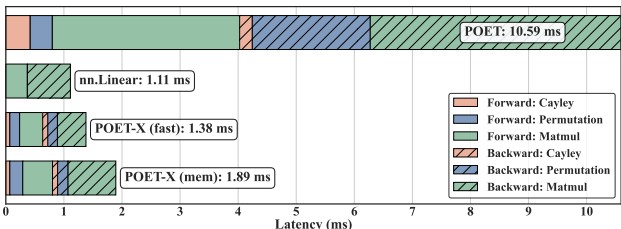

Figure 3. Latency breakdown of POET, POET-X, and PyTorch Linear Layers with sequence length 2048 and block size $b = 256$.

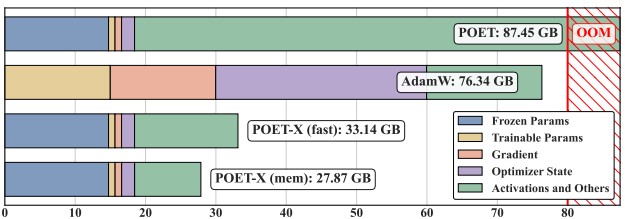

Figure 4. Memory breakdown for training Llama-8B on a single GPU across different methods with batch size 1, sequence lengths 1024, and block size $b = 256$. Since POET (Qiu et al., 2025a) runs OOM under this setting, we estimate its memory footprint by profiling memory usage across different numbers of decoder layers (i.e., parameter sizes) and applying scaling.

precision weights are never stored in memory. For this reason, POET-XQ can be implemented only on POET-X$_{\text{mem}}$, where intermediate activations are recomputed on the fly. In contrast, POET-X$_{\text{fast}}$ would require storing an extra activation tensor, which in turn requires storing high-precision weight matrices. Without custom CUDA kernels, POET-X will need to store high-precision weights and thus cannot support memory-efficient quantized training.

## 4. Experiments and Results

Our experimental validation consists of (1) single-layer profiling to demonstrate the improvements over POET, (2) performance of POET-X on large-scale LLM pretraining, and (3) ablation studies to benchmark the runtime and memory usage by scaling the compute. Experimental settings and additional results are provided in the Appendices.

### 4.1. Single-layer Benchmarking against POET

The original POET (Qiu et al., 2025a) cannot easily scale beyond 3B parameters, since both memory usage and training time become prohibitive. We quantitatively evaluate a single-layer forward and backward pass, showing improvements in both memory footprint and compute time.

We first compare the latency of a single forward and backward pass for POET-X, POET, and a standard linear layer. We set the layer's input/output dimensions and sequence length to match typical settings in Llama-8B. Figure 3 reports the per-operation time breakdown. Compared

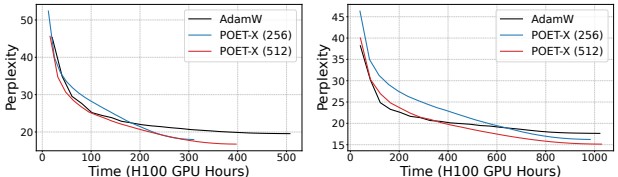

Figure 5. Validation perplexity dynamics with respect to GPU hours for training Llama-8B with $L_{\max} = 256$ (5B tokens) and $L_{\max} = 1024$ (10B tokens), respectively.

| Method | Params (M) | Mem (G) | Val PPL |
|---|---|---|---|
| AdamW | 2764.47 | 81.03 | 12.69 |
| Muon (Kimi) (Liu et al., 2025a) | 2764.47 | 70.94 | **11.45** |
| APOLLO (Zhu et al., 2025) | 2764.47 | 80.60 | 12.97 |
| GaLore (Zhao et al., 2024) | 2764.47 | 74.50 | 14.88 |
| POET-X$_{b=256}$ | 366.64 | **60.58** | 12.76 |
| POET-X$_{b=512}$ | 570.06 | 68.52 | 12.05 |

Table 6. Validation Perplexity (PPL) comparison. The column labeled **Params (M)** reports the total number of trainable parameters for each optimizer, where $b$ is the block size and $L_{\max} = 256$.

| Method | Params (M) | Mem (G) | Val PPL |
|---|---|---|---|
| Quantized 8-bit APOLLO | 2764.47 | 66.37 | 20.49 |
| Quantized 8-bit GaLore | 2764.47 | 66.28 | 17.74 |
| POET-XQ$_{b=256}$ | 366.64 | **51.66** | 16.21 |
| POET-XQ$_{b=512}$ | 570.06 | 60.65 | **14.78** |

Table 7. Validation Perplexity (PPL) comparison. The column labeled **Params (M)** reports the total number of trainable parameters for each optimizer, where $b$ is the block size and $L_{\max} = 256$.

with POET, the total forward and backward time drops from 10.59ms to 1.38ms for POET-X$_{\text{fast}}$ and 1.89ms for POET-X$_{\text{mem}}$. Relative to a highly optimized PyTorch linear layer (cuBLAS), POET-X incurs only modest overhead. Due to higher parameter efficiency, POET-X$_{\text{fast}}$ achieves a backward-pass latency comparable to linear layers.

We profiled the memory consumption for training a Llama-8B model on a single GPU. To ensure a fair comparison, POET, POET-X$_{\text{fast}}$, and POET-X$_{\text{mem}}$ were configured with the exact same number of trainable parameters. A detailed breakdown of memory usage is presented in Figure 4. The visualization highlights distinct memory characteristics for each method. Compared to AdamW, POET should exhibit PEFT-like characteristics by significantly reducing memory consumption for gradients and optimizer states. However, as visualized in the diagram, POET is even worse in terms of memory than AdamW, because its original formulation requires a substantial amount of memory for activations, such as storing the transformed weight matrix ($W_{RP}$) for backpropagation. Consequently, this leads to an overall memory efficiency lower than that of AdamW. In contrast, POET-X$_{\text{fast}}$ and POET-X$_{\text{mem}}$ both exhibit a memory footprint typical of PEFT methods. These characteristics significantly enhance the scalability of the POET-X framework in large-scale pretraining of transformer models.

| Method | Sequence Length 512 | | | | Sequence Length 1024 | | | |
| | $1 \times 1$ H100 | | $1 \times 8$ H100 | | $1 \times 1$ H100 | | $1 \times 8$ H100 | |
| | 8B | 13B | 8B | 13B | 8B | 13B | 8B | 13B |
|---|---|---|---|---|---|---|---|---|
| 8-bit Q-APOLLO | 2.52 | 1.55 | 18.63 | 11.47 | **4.24** | 2.63 | 31.85 | 19.72 |
| 8-bit Q-GaLore | 2.25 | 1.32 | 18.66 | 11.49 | 3.86 | 2.30 | 31.81 | 19.70 |
| POET-XQ$_{b=256}$ | **2.77** | **1.82** | **21.96** | **14.22** | 4.12 | **2.72** | **32.41** | **21.67** |
| POET-XQ$_{b=512}$ | 2.30 | 1.54 | 17.90 | 11.93 | 3.56 | 2.37 | 28.27 | 18.75 |

Table 8. Throughput (k tokens/s) with Llama-8B and Llama-13B.

## 4.2. Mulit-node LLM Pretraining

We evaluate POET-X by pretraining a Llama-3B transformer on the C4 dataset (Raffel et al., 2020), a widely-used, large-scale corpus derived from Common Crawl. We benchmark our method against AdamW, the prevailing optimizer for pretraining, and Muon (Liu et al., 2025a), a recent optimizer that utilizes gradient orthogonalization to maximize training efficiency. We also compare POET against established memory-efficient pretraining methods, including Ga-Lore (Zhao et al., 2024) and APOLLO (Zhu et al., 2025). Experimental settings, including hyperparameters, follow the settings established in (Qiu et al., 2025a; Jaiswal et al., 2024; Liu et al., 2025a). For POET-X, $b$ denotes the block size of the block-diagonal orthogonal matrix.

We design our experiments according to the Chinchilla scaling law (Hoffmann et al., 2022), which suggests an approximate ratio of 20 training tokens per model parameter offers an ideal trade-off between model performance and compute budget. Accordingly, we train a Llama-3B model with a maximum sequence length $L_{max}$ of 256 on 60B tokens to compare different training methods. In Table 6, we observe that POET-X achieves superior validation perplexity (PPL) compared to AdamW and other memory-efficient methods, while slightly underperforming Muon (with much lower GPU memory). Specifically, POET-X$_{b=512}$ yields the second-best validation perplexity of 12.05, offering competitive performance with significantly lower memory.

**Practical training efficiency**. Table 6 shows that POET-X achieves superior iteration-wise convergence, which is consistent with findings on smaller-scale models (Qiu et al., 2025a). To evaluate practical efficiency beyond iteration, we evaluate the wall-clock time convergence in a controlled distributed setting. Experiments were conducted on a multi-node cluster comprising 32 Nvidia H100 GPUs (4 Nodes $\times$ 8 GPUs) connected via InfiniBand. Figure 5 demonstrates that POET-X achieves better wall-clock efficiency than AdamW. The observed efficiency gains in the distributed training stem from the superior memory efficiency of POET-X, which permits the use of the Distributed Data Parallel (DDP), as the entire model, along with all the gradients and optimizer states can fit onto every single GPU, and only data is sharded, which in general provides higher throughput, stronger scalability, and more robustness. However, in the given setting, training AdamW with DDP will lead to OOM. Thus, we choose to use Fully Sharded Data

Parallel (FSDP, with 8 shards and 4 replicates) for training AdamW. FSDP shards the model parameters, gradients, and optimizer states across GPUs, introducing significant collective communication overheads.

**Experiments on POET-XQ**. One of the major advantages of POET-X is its effortless application to quantized models. We conducted pretraining experiments on Llama-3B models trained on 10B tokens; the final validation perplexity is summarized in Table 7. We observe that POET-XQ outperforms both the GaLore and APOLLO baselines, with POET-XQ$_{b=512}$ achieving the overall best performance of 14.78. Notably, POET-XQ achieves these results while maintaining a lower memory footprint. In Table 9, we highlight that POET-XQ$_{b=256}$ requires the overall least training memory. Comparing the computational efficiency, POET-XQ also demonstrates higher throughput (Table 8). This efficiency gain stems largely from the fact that POET-XQ does not optimize low-precision weight matrices directly; this allows it to utilize many standard optimizers, making it easier for POET-X to be used in any quantized base models.

## 4.3. In-depth Efficiency Study

**Memory efficiency**. We investigate whether POET-X's memory efficiency holds when scaling key aspects of pretraining. We benchmark the peak GPU memory during training. This study systematically varies three factors: model size (3B, 8B, and 13B), input sequence length (512, 1024, and 2048), and POET-X's block size $b$ (256 and 512). Memory is measured on a single Nvidia H100 GPU with a batch size of 1. The experiments are performed under two precision settings: a standard setting where all tensors are stored in BF16, and a quantized setting (Zhang et al., 2024; Zhu et al., 2025), where attention and MLP weights are stored in INT8 while gradients and remaining parameters are kept in BF16. We compare POET-X against AdamW, Muon, Ga-Lore, APOLLO, and LoRA. Although LoRA is suboptimal for pretraining (Lialin et al., 2023), it serves as a stringent baseline for memory usage; we therefore match LoRA's trainable parameters to those of POET-X for a fair comparison. Table 9 shows that the original POET is prohibitively memory-intensive and fails to fit 8B and 13B models even at the smallest sequence length. POET-X$_{fast}$ matches the memory efficiency of LoRA, and POET-X$_{mem}$ outperforms all baselines across all scales. This memory advantage is most pronounced at scale. With the 13B model and a 2048 sequence length, POET-X$_{mem,b=256}$ and POET-X$_{mem,b=512}$ consume only 47.21G and 59.02G of memory, respectively.

**Throughput efficiency**. To study POET-X's throughput scalability, we measure the throughput (k tokens/s) across diverse settings, varying the model size, input sequence length, POET-X block size, and the number of nodes. We scale the experiments from 1 to 8 nodes (*i.e.*, from 1 to 64 GPUs). For distributed training, POET-X and LoRA

| Method | Llama-3B | | | Llama-8B | | | Llama-13B | | |
|---|---|---|---|---|---|---|---|---|---|
| | $L_{\max}=512$ | $L_{\max}=1024$ | $L_{\max}=2048$ | $L_{\max}=512$ | $L_{\max}=1024$ | $L_{\max}=2048$ | $L_{\max}=512$ | $L_{\max}=1024$ | $L_{\max}=2048$ |
| AdamW | 28.74 | 28.78 | 31.43 | 78.89 | 76.34 | 78.69 | OOM | OOM | OOM |
| Muon | 18.62 | 19.27 | 21.82 | 50.30 | 53.46 | 54.98 | 76.32 | 77.02 | OOM |
| APOLLO | 19.74 | 20.40 | 21.32 | 51.34 | 53.49 | 56.94 | OOM | OOM | OOM |
| GaLore | 19.06 | 20.17 | 21.16 | 44.52 | 45.62 | 54.71 | 67.15 | 67.86 | 73.37 |
| LoRA$_{r=160}$ | 13.60 | 16.79 | 22.68 | 27.90 | 33.63 | 43.78 | 42.48 | 49.78 | 63.50 |
| LoRA$_{r=320}$ | 16.81 | 19.50 | 25.84 | 33.77 | 38.50 | 49.82 | 50.08 | 57.53 | 71.55 |
| POET$_{b=256}$ | 33.45 | 34.79 | 38.02 | OOM | OOM | OOM | OOM | OOM | OOM |
| POET$_{b=512}$ | 37.11 | 38.09 | 41.34 | OOM | OOM | OOM | OOM | OOM | OOM |
| POET-X$_{\text{fast},b=256}$ | 13.41 | 16.31 | 21.49 | 28.65 | 33.14 | 43.08 | 42.77 | 48.95 | 61.87 |
| POET-X$_{\text{fast},b=512}$ | 17.77 | 20.75 | 26.06 | 36.08 | 41.94 | 50.81 | 53.11 | 59.12 | 71.88 |
| POET-X$_{\text{mem},b=256}$ | **11.96** | **13.28** | **15.47** | **25.94** | **27.87** | **31.74** | **35.65** | **41.62** | **47.21** |
| POET-X$_{\text{mem},b=512}$ | 16.31 | 18.33 | 20.48 | 32.93 | 35.94 | 40.44 | 49.63 | 53.26 | 59.02 |
| 8-bit Q-GaLoRE | 13.76 | 14.49 | **15.78** | 34.25 | 34.89 | 39.29 | 53.79 | 53.38 | 54.82 |
| 8-bit Q-APOLLO | 14.02 | 14.82 | 16.42 | 33.14 | 34.29 | 38.25 | 51.04 | 51.97 | 55.76 |
| POET-XQ$_{b=256}$ | **10.77** | **12.23** | 16.15 | **20.62** | **23.59** | **29.83** | **30.07** | **34.58** | **43.59** |
| POET-XQ$_{b=512}$ | 14.48 | 16.71 | 20.48 | 27.07 | 30.26 | 36.91 | 39.12 | 43.98 | 53.17 |

Table 9. Peak memory (GB) for different models and sequence lengths ($L_{\max}$) on a single H100 (batch size 1, no gradient accumulation).

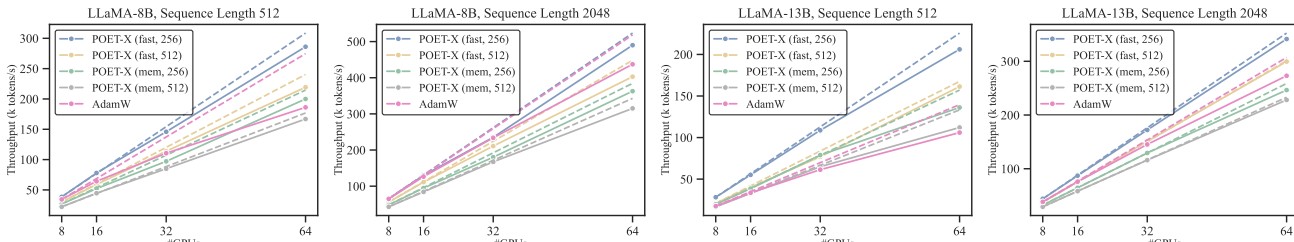

Figure 6. Throughput (k tokens/s) across different numbers of GPUs. The solid line denotes the actual throughput, and the dashed line denotes the ideal linear scaling throughput. The ideal throughput of $k$ GPUs is defined as $T_{k,ideal} = T_{8,real} \times k/8$).

| | Seq. Length 512 (Llama-8B) | | | Seq. Length 2048 (Llama-8B) | | | Seq. Length 512 (Llama-13B) | | | Seq. Length 2048 (Llama-13B) | | |
|---|---|---|---|---|---|---|---|---|---|---|---|---|
| | 1x1 H100 | 8x8 H100 | Ratio | 1x1 H100 | 8x8 H100 | Ratio | 1x1 H100 | 8x8 H100 | Ratio | 1x1 H100 | 8x8 H100 | Ratio |
| AdamW | **7.60** | 186.16 | 24.5 | OOM | 437.21 | N/A | OOM | 105.98 | N/A | OOM | 273.02 | N/A |
| LoRA$_{r=160}$ | 5.28 | **310.59** | **58.8** | 6.42 | 396.97 | 61.8 | 3.68 | **217.52** | 59.1 | 4.42 | OOM | N/A |
| LoRA$_{r=320}$ | 4.03 | 228.80 | 56.8 | 4.82 | 298.41 | **61.9** | 2.81 | 166.78 | 59.3 | 3.26 | OOM | N/A |
| POET-X$_{\text{mem},b=256}$ | 3.73 | 199.95 | 53.7 | 5.92 | 362.75 | 61.2 | 2.74 | 136.14 | 49.7 | 4.02 | 246.47 | 61.3 |
| POET-X$_{\text{mem},b=512}$ | 3.16 | 166.92 | 52.9 | 5.26 | 315.18 | 59.9 | 2.22 | 112.10 | 50.5 | 3.58 | 228.07 | 63.7 |
| POET-X$_{\text{fast},b=256}$ | 5.36 | 286.09 | 53.4 | **8.08** | **489.98** | 60.6 | **3.75** | 206.11 | 55.0 | **5.52** | 341.46 | 61.8 |
| POET-X$_{\text{fast},b=512}$ | 3.84 | 219.30 | 57.2 | 6.96 | 402.88 | 57.9 | 2.69 | 161.33 | **60.0** | 4.65 | 299.46 | **64.3** |

Table 10. Throughput (k tokens/s) comparison between baselines and POET-X. Ratio: throughput ratio between 1x1 H100 and 8x8 H100.

adopt the DDP strategy, and AdamW employs an 8-shard, $N$-replicate hybrid FSDP strategy, where $N$ is the number of nodes. The throughput for single- and 64-GPU runs are presented in Table 10, and the node scaling experiments are shown in Figure 6. In Figure 6, we also compare the empirical throughput (solid line) against the theoretical, ideal linear scaling (dashed line). The ideal scaling of throughput with $k$ GPUs is defined as $T_{k,ideal} = T_{8,real} \times k/8$.

As shown in Table 10 and Table 14 (Appendix), AdamW achieves better single-GPU performance when training Llama-8B with shorter sequence lengths (512 and 1024). However, when scaling either the sequence length or the model size, AdamW encounters OOM errors. Figure 6 also shows that while AdamW's initial training throughput scales well, it soon deviates from the ideal linear scaling curve as the number of nodes increases. This bottle-neck stems from the full-gradient `all-reduce` required across all nodes at every step, which leads to severe network congestion. FSDP's intra-node `all-gather` and `reduce-scatter` operations further reduce throughput. In contrast, POET-X scales well with different model sizes and sequence lengths by minimizing communication overhead with only minimal collective operations.

## 5. Related Work and Concluding Remarks

Recent efforts to improve LLM training efficiency have largely focused on two paradigms: low-rankness and sparsity. Inspired by LoRA (Hu et al., 2022), a significant body of work has explored low-rank structures for efficient pre-training (Lialin et al., 2023; Zhao et al., 2024; Liang et al., 2024; Miles et al., 2024; Huh et al., 2024; Han et al., 2024; Liu et al., 2025b; Zhang et al., 2024; Jaiswal et al., 2024;

Chen et al., 2024; Huang et al., 2025; Su et al., 2025; Huang et al., 2024; Liao et al., 2024). Alternatively, sparsity has been extensively studied to enhance training efficiency (Dao et al., 2019; Hoefler et al., 2021; Chen et al., 2021; Dao et al., 2022; Thangarasa et al., 2023; Qiu et al., 2023; Liu et al., 2024; Qiu et al., 2025a). Falling into this sparse training paradigm, POET-X is a memory- and compute-efficient algorithm, enabling LLM pretraining by effectively scaling orthogonal equivalence transformation. More broadly, there is also a trend of designing stable optimizers for LLMs, such as Muon (Jordan et al., 2024; Liu et al., 2025a), SSO (Xie et al., 2026) and Pion (Shi et al., 2026).

## Impact Statement

This paper presents work whose goal is to advance the field of large-scale efficient training of foundation models. There are many potential societal consequences of our work, none which we feel must be specifically highlighted here.

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

# Appendix

**Table of Contents**

# A. Experimental Details

| Parameter | Llama 3B | Llama 8B | Llama 13B |
|---|---|---|---|
| Hidden dimension | 2560 | 4096 | 5120 |
| Intermediate dimension | 7168 | 14336 | 13824 |
| Number of attention heads | 32 | 32 | 40 |
| Number of hidden layers | 32 | 32 | 40 |

Table 11. Model architectures for different Llama variants.

This section outlines our experimental setup, including the codebase, datasets, and computational resources used.

**Code framework.** Our method is implemented on top of the codebase from (Jaiswal et al., 2024)[1] (Apache 2.0 license), which we also use to train and benchmark the AdamW and LoRA baselines. We will release our code for reproducing all training results prior to publication. The experimental setup consisted of PyTorch 2.8.0 running with CUDA 12.9. We utilized the Hugging Face PEFT[2] code base for all experiments related to LoRA. For a fair comparison, we enable `torch.compile` for all experiments by default.

**Training details.** We utlized the AdamW optimizer (Loshchilov & Hutter, 2017) for updating the POET-related parameters. Detailed hyperparameters for the validation perplexity experiments (Table 6 and Table 7) are provided in Table 12 and Table 13. For the baseline optimizers AdamW, Muon, GaLore, and Apollo, we employed a cosine learning rate scheduler with a decay ratio of 0.1, 5000 warmup steps, a weight decay of 0.01 and, a gradient clipping threshold of 1.0, and a global batch size of 512. For our proposed POET-X, we adopted a minimum learning rate ratio of 0.01.

The hyperparameters of Muon, GaLore and Apollo are taken from their respective repositories. For AdamW, we performed a learning rate grid search over the values $\{1 \times 10^{-2}, 5 \times 10^{-3}, 1 \times 10^{-3}, 5 \times 10^{-4}, 1 \times 10^{-4}\}$ and chose the best-performing value of $\eta_{\text{AdamW}} = 5 \times 10^{-4}$.

Finally, we apply a specific scaling factor $\gamma$ to the learning rate for POET parameters. Since POET optimizes the skew-symmetric matrix $Q$—which transform to the orthogonal matrices $R$ and $P$ via the Cayley-Neumann (approximating $R \approx 2Q$ for small $Q$)—the effective update magnitude differs from that of directly optimizing standard linear layers. Consequently, the POET learning rate is derived directly from the AdamW optimizer's learning rate such that $\eta_{\text{POET}} = \gamma \cdot \eta_{\text{AdamW}}$. We set the scaling coefficient $\gamma = 0.5$ for all experiments.

| Model | Spec. | # GPU | lr (base) | lr (POET) | POET reset gap | training steps | batch size | grad acc. |
|---|---|---|---|---|---|---|---|---|
| Llama 3B | $b = 256$ | 8 | 1e-3 | 5e-4 · $\gamma$ | 400 | 600,000 | 64 | 1 |
| | $b = 512$ | 8 | 1e-3 | 5e-4 · $\gamma$ | 400 | 600,000 | 64 | 1 |

Table 12. Hyper-parameter setup of POET-X.

| Model | Spec. | # GPU | lr (base) | lr (POET) | POET reset gap | training steps | batch size | grad acc. |
|---|---|---|---|---|---|---|---|---|
| Llama 3B | $b = 256$ | 8 | 1e-3 | 5e-4 · $\gamma$ | 400 | 100,000 | 64 | 1 |
| | $b = 512$ | 8 | 1e-3 | 5e-4 · $\gamma$ | 400 | 100,000 | 64 | 1 |

Table 13. Hyper-parameter setup of POET-XQ.

---

[1]https://github.com/jiaweizzhao/GaLore
[2]https://github.com/huggingface/peft

**Experiment details.** For the throughput efficiency experiments, we fix the per-device batch size at 1, adjusting the gradient accumulation steps according to the number of devices. We perform 100 warm-up optimization iterations before measuring the average throughput (tokens/s) across all GPUs over a period of 20 steps. Crucially, all token counts reported in this paper refer to **actual seen tokens**; these metrics exclude padding tokens and therefore cannot be derived solely from the number of training steps, batch size, and sequence length.

**Model architecture.** We implemented the Llama model for pretraining using the **Hugging Face Transformers**[3] library, which is available under the **Apache 2.0** license. The specific architectures for the different Llama model scales are summarized in Table 11. We note that the intermediate dimension of the Feed-Forward Network (FFN) was slightly modified for POET-X from the configurations in (Jaiswal et al., 2024). This adjustment was necessary because the linear layer dimensions are required to be divisible by the POET-X block size, $b$.

**Dataset.** We use the *Colossal Clean Crawled Corpus* (C4) dataset (Raffel et al., 2020) for all pretraining experiments. This dataset is a large-scale, meticulously cleaned version of Common Crawl's web corpus, first introduced for training the T5 model. It has since become a standard dataset for evaluating LLM pre-training algorithms and is available under the **ODC-BY** license.

**Compute Resources.** All the training tasks are performed on **NVIDIA HGX H100 8-GPU System** nodes with 80GB memory each. Depending on the setting, we train on 1, 8, 16, 32 or 64 GPUs. The GPU nodes are interconnected via a fully meshed Infiniband network.

---

[3]https://github.com/huggingface/transformers

# B. Additional Results

**Complete throughput scaling experiments.** We summarize the full throughput scaling experiments in Table 14, with additional results of training Llama-8B and Llama-13B model with a sequence length of 1024. We can observe similar throughput efficiency gain as the longer sequences of 2048, with POET-X$_{\text{fast},b=512}$ exhibiting the best overall throughput ratio for the sequence length 1024.

| Llama-8B | Sequence Length 512 | | | Sequence Length 1024 | | | Sequence Length 2048 | | |
|---|---|---|---|---|---|---|---|---|---|
| | 1x1 H100 | 8x8 H100 | Ratio | 1x1 H100 | 8x8 H100 | Ratio | 1x1 H100 | 8x8 H100 | Ratio |
| AdamW | **7.60** | 186.16 | 24.5 | **9.34** | 277.53 | 29.7 | OOM | 437.21 | N/A |
| LoRA - 160 | 5.28 | **310.59** | **58.8** | 6.20 | 374.87 | 60.5 | 6.42 | 396.97 | 61.8 |
| LoRA - 320 | 4.03 | 228.80 | 56.8 | 4.58 | 273.35 | 59.7 | 4.82 | 298.41 | **61.9** |
| POET-X$_{\text{mem},b=256}$ | 3.73 | 199.95 | 53.7 | 5.29 | 294.25 | 55.6 | 5.92 | 362.75 | 61.2 |
| POET-X$_{\text{mem},b=512}$ | 3.16 | 166.92 | 52.9 | 4.42 | 261.32 | 59.2 | 5.26 | 315.18 | 59.9 |
| POET-X$_{\text{fast},b=256}$ | 5.36 | 286.09 | 53.4 | 7.11 | **438.93** | 61.7 | **8.08** | **489.98** | 60.6 |
| POET-X$_{\text{fast},b=512}$ | 3.84 | 219.30 | 57.2 | 5.56 | 349.71 | **62.9** | 6.96 | 402.88 | 57.9 |

| Llama-13B | Sequence Length 512 | | | Sequence Length 1024 | | | Sequence Length 2048 | | |
|---|---|---|---|---|---|---|---|---|---|
| | 1x1 H100 | 8x8 H100 | Ratio | 1x1 H100 | 8x8 H100 | Ratio | 1x1 H100 | 8x8 H100 | Ratio |
| AdamW | OOM | 105.98 | N/A | OOM | 177.63 | N/A | OOM | 273.02 | N/A |
| LoRA - 160 | 3.68 | **217.52** | 59.1 | 4.14 | 255.24 | 61.7 | 4.42 | OOM | N/A |
| LoRA - 320 | 2.81 | 166.78 | 59.3 | 3.14 | 188.86 | 60.1 | 3.26 | OOM | N/A |
| POET-X$_{\text{mem},b=256}$ | 2.74 | 136.14 | 49.7 | 3.64 | 212.17 | 58.3 | 4.02 | 246.47 | 61.3 |
| POET-X$_{\text{mem},b=512}$ | 2.22 | 112.10 | 50.5 | 3.03 | 174.46 | 57.6 | 3.58 | 228.07 | 63.7 |
| POET-X$_{\text{fast},b=256}$ | **3.75** | 206.11 | 55.0 | **4.90** | **304.06** | 62.1 | **5.52** | **341.46** | 61.8 |
| POET-X$_{\text{fast},b=512}$ | 2.69 | 161.33 | **60.0** | 3.84 | 242.01 | **63.0** | 4.65 | 299.46 | **64.3** |

Table 14. Throughput (k tokens/s) comparison between baselines and POET-X. Ratio: throughput ratio between 1x1 H100 and 8x8 H100.

**Complete throughput scaling curves.** We additionally provide the comparison of ideal linear scaling throughput and the actual scaling throughput for the sequence length of 1024 in Figure 7.

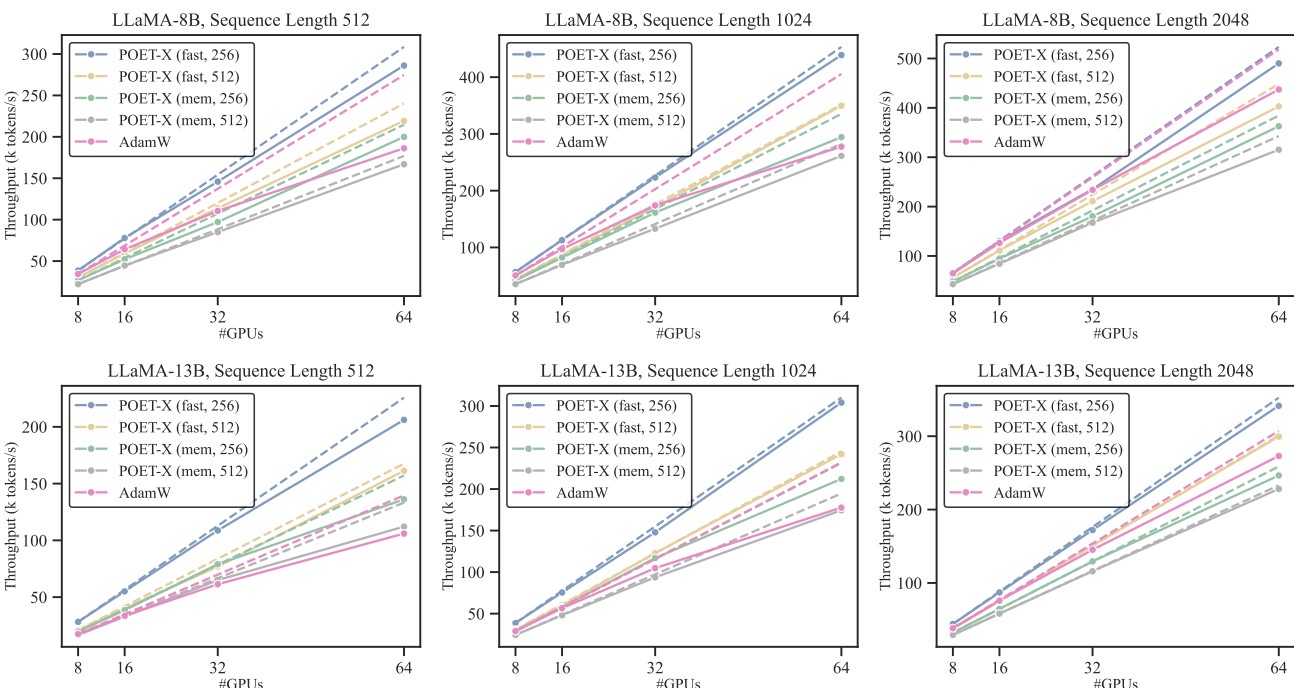

Figure 7. Throughput (k tokens/s) across different numbers of GPUs. The solid line denotes the actual throughput, and the dashed line denotes the ideal linear scaling throughput. The ideal throughput of $k$ GPUs is defined as $T_{k,ideal} = T_{8,real} \times k/8$).

**Additional experimental results on Qwen3 architectures.** To further evaluate the architectural generalization of POET-X, we conducted additional experiments on the Qwen3 architecture. Specifically, we trained a 1B Qwen3 dense model and a 1B Qwen3 Mixture-of-Experts (MoE) model. To ensure fair comparison and adhere to the Chinchilla-optimal token budgets, both models were trained on 20B tokens.

The results, along with their respective memory usage, are summarized in Table 15. We observe consistent performance gains with POET-X on the Qwen3 architecture. Interestingly, the performance improvement in terms of validation perplexity is even more pronounced in the MoE structure.

| Architecture | Method | Val PPL ($\downarrow$) | Memory (GB) ($\downarrow$) |
|---|---|---|---|
| Qwen3 1B | AdamW | 14.78 | 36.09 |
|  | POET-X | **14.67** | **29.42** |
| Qwen3 MoE 1B | AdamW | 16.89 | 28.41 |
|  | POET-X | **15.38** | **27.51** |

Table 15. Validation perplexity and memory usage for Qwen3 1B dense and MoE models trained on 20B tokens. Best results are bolded.

