# OpenReview forum: "POET-X: Memory-efficient LLM Training by Scaling Orthogonal Transformation"
_ICML.cc/2026/Conference — ICML 2026 spotlight_

### Official Review · Reviewer_BEPy · 2026-02-27

**Soundness:** 3
**Presentation:** 3
**Significance:** 4
**Originality:** 3
**Overall Recommendation:** 5
**Confidence:** 4

**Summary:**

The paper introduces POET-X, a scalable and memory-efficient variant of the Reparameterized Orthogonal Equivalence Training (POET) algorithm designed for large language model (LLM) pretraining. While the original POET framework provides strong training stability via spectrum preservation, its weight-centric formulation requires intensive matrix multiplications that lead to prohibitive memory consumption and computational overhead. POET-X resolves this by reformulating the computations into an input-centric sequence of linear maps, effectively eliminating the need to store large intermediate activations associated with weight matrices. Furthermore, the authors propose a batch-parallel strategy for block-diagonal matrix multiplications and optimize the Cayley-Neumann Parameterization (CNP) by storing only the upper-triangular portion of skew-symmetric matrices. These improvements are implemented via custom Triton/CUDA kernels. Empirically, POET-X enables the pretraining of a 13B parameter LLM on a single NVIDIA H100 GPU , achieving a 3x reduction in memory and an 8x speedup compared to the original POET , while maintaining validation perplexity competitive with or superior to AdamW.

**Compliance With Llm Reviewing Policy:**

Affirmed.

**Final Justification:**

The rebuttal did not change my evaluation but strongly reinforced my initial positive assessment. I have no remaining reservations and confidently support the acceptance of this work.

**Key Questions For Authors:**

1. Generalizability and Hardware Coupling: POET-X heavily relies on customized Triton and CUDA kernels, particularly for the fused Cayley-Neumann parameterization. How easily can this input-centric formulation and operator fusion be adapted to non-NVIDIA hardware, or applied to non-standard model architectures (e.g., Mixture-of-Experts) that involve dynamic routing?

2. Hyperparameter Sensitivity: The appendix outlines a specific learning rate scaling factor ($\gamma=0.5$) and a complex transient gradient clipping schedule (clipping norm scaled from 0.01 to 1.0 over the first 2000 steps). How sensitive is POET-X to these specific hyperparameters? Does the algorithm require extensive tuning when migrating to different datasets or model scales compared to standard AdamW?

3. Long-term Truncation Error: The $k=3$ truncation in CNP sacrifices strict orthogonality. Over much longer pretraining horizons (e.g., 1T+ tokens) than the 60B tokens tested, does this non-orthogonality error compound? Have you observed hyperspherical energy divergence or late-stage training instability during extended runs?

4. Capacity vs. Efficiency Trade-off: Table 6 shows Muon achieving a lower validation PPL (11.45) than $POET-X_{b=512}$ (12.05). Does forcing the parameters into a block-diagonal orthogonal constraint inherently bottleneck the representational capacity of the model compared to Muon's approach? Under what specific resource constraints should a practitioner explicitly choose POET-X over Muon?

**Limitations:**

No. While the authors discuss the compute-memory trade-off between the $POET-X_{fast}$ and $POET-X_{mem}$ variants, they have not adequately discussed the broader limitations of the framework. The paper lacks a dedicated Limitations section addressing the potential long-term negative impacts of the CNP truncation error and the significant engineering overhead required to maintain custom Triton kernels  for future model architectures. Furthermore, an honest discussion regarding the apparent capacity gap when compared to Muon (as seen in the perplexity results)  is missing. Adding a section to critically evaluate these points would greatly strengthen the submission.

**Strengths And Weaknesses:**

Soundness: The algorithmic transformations—specifically the input-centric reformulation, permutation reduction, and batch-wise block-diagonal multiplication—are mathematically well-founded and effectively address the $\mathcal{O}(nm^2)$ complexity bottleneck of the original method. The empirical evaluation is rigorous, scaling up to 64 GPUs and 13B parameters, adhering to Chinchilla scaling laws for the 60B token runs. However, a notable weakness exists regarding the truncation of the Cayley-Neumann series ($k=3$), which the authors acknowledge trades strict orthogonality for numerical efficiency. The long-term compounding of this truncation error in industry-standard pretraining runs (e.g., $>1$T tokens) is not theoretically bounded or empirically tested. Additionally, the paper claims POET-X avoids sacrificing performance, yet $POET-X_{mem}$ explicitly trades compute time for memory via activation recomputation , and the model's representational capacity appears slightly bottlenecked compared to Muon (Validation PPL 12.05 vs. 11.45).

The submission is well-structured, and the narrative flow detailing the transition from POET's bottlenecks to POET-X's specific hardware-aware solutions is logical. However, the data visualization could be improved for better accessibility. Specifically, the color palette used in Figure 6 (throughput scaling across different GPU counts) lacks sufficient contrast. As a reader with mild color vision deficiency, I found it difficult to distinguish between the overlapping dashed and solid lines for $POET-X_{fast}$, $POET-X_{mem}$, and AdamW. I recommend adopting a palette with higher contrast or incorporating distinct shape markers to clearly convey the performance deltas.

Significance: Addressing the prohibitive memory requirements of full-parameter LLM pretraining is a critical problem. Enabling billion-parameter pretraining on single consumer-accessible nodes  provides high practical utility to the community. The native support for INT8 quantized training (POET-XQ) further amplifies its impact. The primary weakness here is that the high dependency on customized Triton/CUDA kernels  may tightly couple the method to NVIDIA hardware, potentially hindering broad adoption across different AI accelerators or novel non-standard architectures (e.g., MoE, State Space Models) compared to framework-native optimizers.

Originality: The work offers a highly original systems-algorithm co-design. While concepts like matrix-free linear systems and gradient checkpointing exist, combining them with a specialized, memory-halving Cayley-Neumann parameterization and custom kernel fusion to solve the specific $\mathcal{O}(nm^2)$ bottleneck of orthogonal equivalence transformations  represents a novel and substantial engineering contribution to the field of sparse training.

---

> ### Author Rebuttal · Authors · 2026-03-31
>
> We sincerely thank the reviewer for the constructive comments on our work. We take every comment seriously and hope our response can address the reviewer’s concerns.
>
> ---
>
> **Q1: Specifically, the color palette used in Figure 6 lacks sufficient contrast**
>
> A1: Thanks for the suggestion. We will update Figure 6 to incorporate a high-contrast color palette.
>
> ---
>
> **Q2: Generalizability & Hardware Coupling: POET-X relies on customized CUDA kernels. How easily can it be adapted to non-NVIDIA hardware, or applied to MoE?**
>
> A2: Great question! While currently implemented in Triton for NVIDIA hardware, POET-X’s Cayley-Neumann parameterization is mathematically hardware-agnostic. One advantage of using Triton is its support for multiple hardware backends, enabling direct porting of our custom kernels to AMD (ROCm) or Intel (XPU).
>
> While the present study focuses on the Llama architecture, POET-X is fundamentally designed to operate on **linear layers**, which is the core of most Transformer-based models. MoE extends the Llama architecture by replacing each FFN with multiple expert FFNs and a routing gate, both of which are linear layers. Consequently, POET-X is natively applicable to MoE architectures in the same manner. We will include additional experiments on other architectures in the revision.
>
> ---
>
> **Q3: How sensitive to hyperparameter is this method, does it require extensive tuning? E.g., the method uses a transient gradient clipping schedule for the first 2000 steps.**
>
> A3: Thanks for the question. First of all, gradient clipping (GC) is used by default for all training methods, as this is a common practice for LLM pretraining. To further address the reviewer’s concern, we added an ablation study training a 3B model for 50,000 steps. The results indicate that standard fixed-norm clipping provides equal stability, and our specialized clipping scheme (stricter clipping for the first 2,000 steps) is not strictly necessary for convergence. We retained this clipping to match the original POET for a fair comparison.
>
> ***Impact of GC***
>
> | Method | Val PPL |
> | :--- | :--- |
> | Standard GC | 18.275 |
> | POET paper GC | 18.274  |
>
> ---
>
> **Q4: The truncation in CNP sacrifices orthogonality. Have you observed late-stage training instability during extended runs as the error compound?**
>
> A4: Great question. It is exactly why POET performs this merge-then-reinitialize step. The Neumann series converges when the operator norm of $Q$ is less than 1, and this reinitialization trick can ensure the Neumann series stays at a great precision (since $Q$ starts at 0). This guarantees that the orthogonality error does not explode over the trillion of tokens, as we constantly reset $R$ and $P$ to identity matrix. Empirically we did not observe any training instabilities or any divergence of hyperspherical energy. We also added the experiment, where we double the Chinchilla optimal tokens, by training a Llama 1B model on 40B tokens to evaluate POET-X in the over-training regime. As shown below, POET-X still outperforms AdamW with great stability.
>
> ***Over-Training Evaluation ($G=40$)***
>
> | Method | Val PPL |
> | :--- | :--- |
> | AdamW | 13.57 |
> | POET-X | **13.31**  |
>
> ---
>
> **Q5: Capacity vs. Efficiency: Table 6 shows Muon achieving a lower PPL. Under what specific resource constraints should a practitioner choose POET-X?**
>
> A5: Great question! We note that POET-X represents a novel, spectrum-preserving paradigm with significant potential for further refinement. Importantly, the Muon baseline utilizes the **highly optimized Kimi implementation**, which incorporates several critical improvements over the original version of Muon to achieve the reported performance:
>
> - **Weight decay:** identified by Kimi as critical for scaling stability.
> - **Calibrated learning rate:** a manually tuned 0.2 scaler to align the RMSNorm updates with AdamW.
> - **Hybrid optimization:** selective application of Muon or AdamW to specific layers.
> - **Distributed momentum buffer:** necessary to make the otherwise prohibitive computational overhead of Muon trainable.
>
> By matching the performance of Muon without such extensive tuning, POET-X establishes a natively scalable and efficient framework for large-scale LLM training.
>
> More importantly, POET-X offers superior memory efficiency over Muon while uniquely enabling large-scale pretraining in low-precision regimes, making it a compelling solution for practitioners operating in strictly memory-constrained environments.
>
> ---
>
> **Q6: The paper lacks a Limitations section and an honest discussion regarding the capacity gap when compared to Muon.**
>
> A6:  Great suggestion! We will include a Limitations section to include:
>
> - **Architectural Scope:** Evaluation currently centers on Llama; expanding to other architectures remains important.
>
> - **Optimization Maturity:** As a new paradigm, POET-X lacks the extensive engineering like AdamW or Muon, explaining the small performance gap compared to the highly tuned Muon.

---

> > ### Author Rebuttal · Reviewer_BEPy · 2026-04-03
> >
> > I thank the authors for the clear and effective rebuttal. It has fully resolved all my initial concerns, and I have no remaining doubts about the work. Therefore, I will maintain my original score.

---

> > > ### Author Response · Authors · 2026-04-04
> > >
> > > We are very happy that we were able to address all of your concerns and want to sincerely thank the reviewer for their time, rigorous evaluation, and constructive feedback throughout this process.

---

### Official Review · Reviewer_srpv · 2026-03-10

**Soundness:** 3
**Presentation:** 3
**Significance:** 3
**Originality:** 3
**Overall Recommendation:** 5
**Confidence:** 4

**Summary:**

This paper proposes POET-X, a scalable and memory-efficient variant of POET for large language model training. The original POET optimizes weight matrices through orthogonal equivalence transformations that preserve the spectrum of the weight matrix, providing training stability. However, POET suffers from prohibitive memory consumption and slow runtime due to intensive matrix multiplications. POET-X addresses these limitations through several technical contributions: (1) an input-centric reformulation that replaces expensive matrix-matrix multiplications with matrix-vector multiplications, (2) customized CUDA kernels for efficient permutation operations with permutation reduction, (3) a batch-parallel strategy for block-diagonal matrix multiplications that avoids constructing full sparse matrices, (4) an efficient Cayley-Neumann parameterization using upper-triangular storage and Triton kernel fusion, and (5) gradient checkpointing for further memory savings. The paper also introduces POET-XQ, a quantized variant for even lower memory usage. Experiments on Llama-3B/8B/13B models show that POET-X achieves 3× GPU memory reduction and 8× runtime speedup over POET, with memory efficiency comparable to LoRA and performance better than AdamW.

**Compliance With Llm Reviewing Policy:**

Affirmed.

**Final Justification:**

The newly added Qwen3 dense and Qwen3 MoE experiments effectively address my concerns about the method’s generalization across different architectures, with the results on the MoE architecture being reasonably supportive. In addition, the authors provided experiments under different training-step budgets, as well as more detailed visualizations of the raw training dynamics. These results are informative and further strengthen the credibility of the paper’s conclusions. The authors have addressed all my previous concerns. Overall, I think this is a good paper, and I am raising my score from 4 to 5.

**Key Questions For Authors:**

See the detailed weaknesses above.

**Limitations:**

The authors should include a dedicated Limitations section or append one to the conclusion.

**Strengths And Weaknesses:**

>**Disclosure (Policy B):** In compliance with the ICML 2026 LLM Policy B (Permissive), to improve linguistic accuracy and communication efficiency, I used a LLM to translate and polish this review. All opinions, technical analyses, and judgments were written by me in my native language; the LLM was used only for translation and language refinement.

**Strengths:**

The work focuses on a practically important problem: enabling memory-efficient pretraining of billion-parameter LLMs. The demonstrated ability to pretrain Llama-8B on a single H100 GPU (where AdamW runs OOM) is a notable practical achievement. POET-X achieves better perplexity than AdamW with LoRA-level memory footprint during pretraining (not just finetuning), which is a compelling result. The distributed training results showing that POET-X can use DDP instead of FSDP due to its memory efficiency, leading to better throughput scaling, highlight a practical benefit.

**Weaknesses:**

- The algorithmic robustness is questionable due to the reliance on fragile heuristics. For instance, the authors must implement a highly specific "transient gradient clipping schedule" (clipping norm of 0.01 linearly increased to 1.0 over 10 steps) to stabilize the first 2000 steps of training. This suggests underlying optimization instabilities that are not well addressed.

- The training dynamics curve is missing. It is difficult to assess the stability and convergence behavior of POET-X compared to baselines. The paper should include training curves (e.g., training loss vs. steps) to provide insights into the optimization process and justify the need for such heuristics.

- Despite the claims of superiority, POET-X still underperforms the Muon baseline in terms of validation perplexity (12.05 vs. 11.45), and the paper lacks a fundamental analysis of why this performance gap exists.

- The paper focuses exclusively on the Llama architecture. Generalization to other architectures (e.g., Mixture-of-Experts) is not discussed.

- The experiments strictly follow Chinchilla-optimal token budgets. Since modern LLMs rely heavily on over-training to reduce downstream inference costs, it remains unclear whether POET-X maintains its stability and competitive perplexity scaling under massive over-training regimes.

- Section 5 reads like a mere citation dump. Lumping numerous concurrent works into single sentences without critical comparison fails to adequately contextualize POET-X's specific contributions. A more rigorous discussion is required.

- **Formatting Violation**: Throughout the manuscript, table captions are incorrectly placed below the tables instead of above them. This reflects poorly on the submission's polish and professionalism.

---

> ### Author Rebuttal · Authors · 2026-03-31
>
> We sincerely thank the reviewer for the constructive comments on our work. We take every comment seriously and hope our response can address the reviewer’s concerns.
>
> ---
>
> **Q1: The algorithmic robustness is questionable due to the reliance on fragile heuristics, e.g., the "transient gradient clipping schedule" to stabilize the first 2000 steps of training.**
>
> Thanks for the question. First of all, gradient clipping is used by default for all training methods, as this is a common practice for LLM pretraining. To further address the reviewer’s concern, we added an ablation study training a 3B model for 50,000 steps. The results indicate that standard fixed-norm clipping provides equal stability, and our specialized clipping scheme (stricter clipping for the first 2,000 steps) is not strictly necessary for convergence. We retained this clipping to match the original POET for a fair comparison, though we later found it unnecessary.
>
> ***Impact of Gradient Clipping (GC) on Training Stability and Performance***
>
> | Method | Val PPL |
> | :--- | :--- |
> | Standard GC | 18.275 |
> | POET paper GC | 18.274  |
>
> ---
>
> **Q2: The paper should include training loss curves to provide insights into the optimization process.**
>
> Great suggestion! We will add the training curves to the revision.
>
> Examples: https://anonymous.4open.science/r/icml2026_poetx_rebuttal-7108/README.md
>
> ---
>
> **Q3: POET-X still underperforms Muon in perplexity**
>
> Great question! We note that POET-X represents a novel, spectrum-preserving paradigm with significant potential for further refinement. Importantly, the Muon baseline utilizes the **highly optimized Kimi implementation**, which incorporates several critical improvements over the original version of Muon to achieve the reported performance:
>
> - **Weight decay:** identified by Kimi as critical for scaling stability.
> - **Calibrated learning rate:** a manually tuned 0.2 scaler to align the RMSNorm updates with AdamW.
> - **Hybrid optimization:** selective application of Muon or AdamW to specific layers.
> - **Distributed momentum buffer:** necessary to make the otherwise prohibitive computational overhead of Muon trainable.
>
> By being comparable to the performance of Muon without using any complicated strategy and extensive hyperparameter tuning, POET-X establishes a more natively scalable and efficient framework for large-scale LLM training.
>
> More importantly, POET-X offers much stronger memory efficiency over Muon while POET-X can also effortlessly enable large-scale pretraining in low-precision regimes (this is highly nontrivial for Muon), making it a compelling solution for practitioners operating in strictly memory-constrained environments.
>
> ---
>
> **Q4: The paper focuses exclusively on Llama.**
>
> Thank you for the question. While the present study focuses on the Llama architecture, POET-X is fundamentally designed to operate on **linear layers**, which is the core of most Transformer-based models. MoE extends the Llama architecture by replacing each FFN with multiple expert FFNs and a routing gate, both of which are linear layers. Consequently, POET-X is natively applicable to MoE architectures in the same manner. In practice, we don’t find it a problem for POET-X to run on other LLM architectures. We will supplement additional results on other architectures in our final version.
>
> ---
>
> **Q5: The experiments strictly follow Chinchilla-optimal token budgets and is unclear how POET-X performs under massive over-training regimes.**
>
> Excellent question! We conducted all the experiments following the Chinchilla scaling law as it provides a standardized, fair setting for comparison. To evaluate POET-X in the over-training regime, we trained a 1B model on 40B tokens ($G=40$), doubling the Chinchilla-optimal budget. As shown below, POET-X outperforms the AdamW baseline. This result confirms that POET-X remains stable and superior in over-trained scenarios.
>
> ***Over-Training Evaluation ($G=40$)***
>
> | Method | Val PPL |
> | :--- | :--- |
> | AdamW | 13.57 |
> | POET-X | **13.31**  |
>
> ---
>
> **Q6: Section 5 fails to adequately contextualize POET-X's specific contributions. A more rigorous discussion is required.**
>
> Great suggestion! We will expand Section 5 to include a detailed comparison with existing works in our revision. We will highlight POET-X’s unique memory efficiency and suitability for large-scale, low-precision distributed pretraining.
>
> ---
>
> **Q7: The table captions are incorrectly placed below the tables.**
>
> We will fix these errors in the revised version.
>
> ---
>
> **Q8: The authors should include a dedicated Limitations section.**
>
> Great suggestion! We will include a Limitations section:
>
> - **Architectural Scope:** Evaluation currently centers on Llama; expanding to MoE and other architectures remains important.
>
> - **Optimization Maturity:** As a new paradigm, POET-X lacks the extensive engineering of mature baselines like AdamW or Muon, likely explaining the small performance gaps compared to highly tuned Muon.

---

> > ### Author Rebuttal · Reviewer_srpv · 2026-04-02
> >
> > Thank you for the rebuttal. I appreciate the detailed responses and hope my comments can help improve the paper. That said, I still have two follow-up concerns that I do not think have been fully resolved.
> >
> > - **Claim regarding MoE / generalization**：The rebuttal argues that POET-X operates on linear layers and therefore can, in principle, be applied to the expert FFNs and routing gates in MoE models. This is reasonable at the operator level, since the method is built around orthogonal equivalence transformations of linear layers. However, this does not sufficiently support generalization at the architecture level. The empirical evaluation is still almost entirely limited to Llama-style dense models. The paper provides no experiments on MoE or other non-Llama architectures, nor any analysis of whether the claimed benefits in memory, throughput, and stability would persist under the sparse routing, load imbalance, and different communication patterns of MoE training. In other words, the rebuttal shows formal applicability to linear layers, but not actual validation on MoE or preservation of the claimed benefits there. For this reason, I do not think it is justified to state so strongly that “it is not a problem for POET-X to run on other LLM architectures.”
> > - **Concern regarding training dynamics / stability**：Both the submission and the rebuttal present training stability as a central strength, but the evidence remains focused mainly on final perplexity, memory usage, throughput, and scaling, rather than directly showing that the optimization process itself is stable. Although the authors shared training curves in the anonymous link, they seem to cover only a single setting, with unclear model scale and training configuration, and they appear to be smoothed. Smoothed curves cannot adequately reveal early-stage oscillation, loss spikes, or other transient instabilities. As a result, I do not think these plots are sufficient to strongly support the paper’s stability claims. **Providing raw, unsmoothed training dynamics for already completed runs, such as loss, perplexity, and possibly gradient-norm curves, does not seem unrealistic at the rebuttal stage.** Moreover, from the curves provided, POET-X appears to lag behind the AdamW baseline when the number of optimization steps is limited. Therefore, I remain unconvinced about its early-stage stability and performance, especially for larger-scale models.
> >
> > Based on these concerns, I will maintain my current score.

---

> > > ### Author Response · Authors · 2026-04-04
> > >
> > > We sincerely thank the reviewer for these additional comments on our work. We take every comment seriously and hope our response can address the reviewer’s concerns.
> > >
> > > ---
> > >
> > > **Q1. How can you claim POET-X generalizes to other LLM architectures without providing empirical validation on Mixture-of-Experts (MoE) or non-Llama models? Will the claimed benefits in memory, throughput, and stability actually persist under MoE-specific conditions like sparse routing, load imbalance, and different communication patterns?**
> > >
> > > A1: We appreciate the reviewer’s valuable suggestion. To directly address this concern of architectural generalization, we took some time and conducted additional experiments on other LLM architectures (hence the late reply). Our additional experiments include training a 1B Qwen3 dense model and a 1B Qwen3 MoE model. Both models were trained on 20B tokens to match the Chinchilla-optimal token budgets. The results of these runs, along with their respective memory usage, are summarized in the table below:
> > >
> > >
> > >  ***Qwen3 1B (Training on 20B tokens)***
> > >
> > > | Method | Val PPL | Memory (G) |
> > > | :--- | :--- | :--- |
> > > | AdamW | 14.78 | 36.09 |
> > > | POET-X | **14.67** | **29.42** |
> > >
> > >  ***Qwen3 MoE 1B (Training on 20B tokens)***
> > >
> > > | Method | Val PPL | Memory (G) |
> > > | :--- | :--- | :--- |
> > > | AdamW | 16.89 | 28.41 |
> > > | POET-X | **15.38**  | **27.51** |
> > >
> > > We want to highlight that due to the limited time constraint of the rebuttal phase, we **did not** perform any hyperparameter tuning but instead simply re-used the same set of hyperparameters from the 1B Llama architecture experiments. We can still see the same performance gain, interestingly, in the MoE structure, the performance gain is actually bigger. Due to the time constraint, the current version is directly adapted from dense models without any modifications, and the memory-efficiency of POET-X can be further optimized for MoE.
> > >
> > > ---
> > >
> > > **Q2 Why does POET-X appear to lag behind the AdamW baseline when optimization steps are limited, and what does this imply for early-stage stability in larger-scale models?**
> > >
> > > A2: Great question! The training loss trajectory is highly dependent on the specific learning rate scheduler employed by each method. As discussed in the paper, POET-X inherently exhibits a slower initial loss reduction compared to AdamW, which is then followed by a phase of smooth, consistent decrease. To directly address the reviewer’s concern regarding POET-X's stable gain under limited training budgets, we have conducted two additional experiments constrained to 50,000 and 100,000 steps. As shown in the table below, these results demonstrate that POET-X remains highly effective and the gain is quite stable even with much fewer training steps. But we want to highlight that POET-X’s performance gain is more present for longer training runs and with bigger base models.
> > >
> > >  ***PPL Comparison in under-trained Regimes***
> > >
> > > | Method | Val PPL |
> > > | :--- | :--- |
> > > | AdamW (50K) | 16.36 |
> > > | POET-X (50K) | **16.17** |
> > > | AdamW (100K) | 15.03 |
> > > | POET-X (100K) | **14.72** |
> > >
> > > ---
> > >
> > > **Q3. Can you provide raw, unsmoothed training dynamics to prove actual optimization stability, since smoothed curves hide transient instabilities like early-stage oscillations?**
> > >
> > > A3: Thanks for the question! We agree with the reviewer that training dynamics are informative. To address the reviewer’s concern, we have provided the raw training loss and validation perplexity curves for the Llama 3B (trained on 5B, 10B, and 60B tokens), Qwen3 1B (20B tokens), and Qwen3 MoE 1B (20B tokens) models. We will ensure these figures are included in the revised manuscript. As demonstrated in the plots, the raw curves show no dramatic early-stage oscillations or loss spikes and remain stable throughout the entire training stage.
> > >
> > > Llama 3B (5B):
> > > https://anonymous.4open.science/r/icml2026_rebuttal_second_round-DB3C/llama_3b_50000.pdf
> > >
> > > Llama 3B (10B):
> > > https://anonymous.4open.science/r/icml2026_rebuttal_second_round-DB3C/llama_3b_100000.pdf
> > >
> > > Llama 3B (60B):
> > > https://anonymous.4open.science/r/icml2026_rebuttal_second_round-DB3C/llama_3b_600000.pdf
> > >
> > > Qwen3 1B (20B):
> > > https://anonymous.4open.science/r/icml2026_rebuttal_second_round-DB3C/qwen3_200000.pdf
> > >
> > > Qwen3 MoE 1B (20B):
> > > https://anonymous.4open.science/r/icml2026_rebuttal_second_round-DB3C/qwen3_moe_200000.pdf

---

### Official Review · Reviewer_kKuH · 2026-03-13

**Soundness:** 3
**Presentation:** 2
**Significance:** 2
**Originality:** 2
**Overall Recommendation:** 3
**Confidence:** 2

**Summary:**

This paper proposes POET-X, a fast, scalable, and memory-efficient variant of POET for LLM training by making **orthogonal equivalence transformations** practical at scale.

The authors introduce several implementation strategies:
(i) rewriting POET’s weight-centric update as an **input-centric** sequence of linear maps,
(ii) accelerating and simplifying permutation operations via custom CUDA/Triton implementations,
(iii) exploiting block-diagonal structure through batch-parallel multiplications,
(iv) improving the storage and computation efficiency of the Cayley-Neumann parameterization, and
(v) using checkpointing to further reduce activation memory.

Experimentally, they report substantial single-layer speedups, significant end-to-end memory savings, and competitive results on LLM pretraining.

**Compliance With Llm Reviewing Policy:**

Affirmed.

**Key Questions For Authors:**

NA

**Strengths And Weaknesses:**

**Strengths**

* **Clear system-level scaling story for POET.** The paper identifies the main computational and memory bottlenecks in POET, including activations, permutation operations, block-structured multiplications, and the Cayley-Neumann parameterization, and addresses them with concrete engineering solutions.

* **The core reformulation is well motivated.** The input-centric computation rewrites the original weight-centric update into a sequence of three matvec-style operations, directly targeting the activation and memory overhead that made the original POET difficult to scale in LLM training.

* **Strong single-layer profiling evidence.** The latency breakdown provides convincing evidence that POET-X substantially reduces the forward-and-backward runtime relative to the original POET implementation.

* **Substantial memory savings.** The memory breakdown shows significant reductions compared with AdamW and also large improvements over the profiled POET setting, supporting the paper’s claim that POET-X is much more memory efficient in practice.

* **Reasonably broad end-to-end pretraining comparisons.** The experimental section includes several relevant baselines, such as Muon, APOLLO, GaLore, and LoRA, and reports validation perplexity together with the number of trainable parameters, which helps contextualize the efficiency-performance trade-off.

**Weaknesses**

* **The scientific novelty is primarily at the systems and implementation level.** While the engineering contribution is clear, the paper offers limited new insight from a learning-theoretic perspective. The claims regarding stability and effectiveness remain largely empirical.

* **There may be some reproducibility risk.** The reported gains appears to rely on specialized CUDA/Triton kernels and carefully fused implementations. Without open-sourcing these kernels and providing detailed build and runtime instructions, it may be difficult to reproduce and validate the results.

* **Some key ablations feel incomplete.** The paper should include important design choices, such as the block size, the Neumann truncation level, orthogonality error vs training performance, and the recomputation cost of the memory-efficient variant.

* **Some headline claims need more complete experimental details.** Statements such as “pretraining up to 13B on a single H100” would be more convincing if accompanied by a fuller specification of the training setup.

---

> ### Author Rebuttal · Authors · 2026-03-31
>
> We sincerely thank the reviewer for the constructive comments on our work. We take every comment seriously and hope our response can address the reviewer’s concerns.
>
> ---
>
> **Q1: The scientific novelty is primarily at the systems and implementation level. While the engineering contribution is clear, the paper offers limited new insight from a learning-theoretic perspective.**
>
> Our contribution lies primarily in the novel application of numerical linear algebra and system-level CUDA kernel optimization to achieve stable, efficient LLM pretraining. Although the work involves substantial engineering, the outcome is significant: POET-X becomes one of the most memory-efficient pretraining algorithms, with performance surpassing AdamW.
>
> Moreover, our scalable way for optimizing large orthogonal matrices may also be of independent interest for the machine learning community.
>
> ---
>
> **Q2: Because of the specialized CUDA/Triton kernels and carefully fused implementations, open-sourcing these kernels is important to reproduce the results.**
>
> Great suggestion! We promise to fully release the source code, including the training code, all the kernel implementations and all the training scripts for reproducing the results.
>
> ---
>
> **Q3: Some key ablations feel incomplete, such as the block size, the Neumann truncation level, orthogonality error vs training performance, and the recomputation cost of the memory-efficient variant.**
>
> Great suggestion! We have added the following ablation studies to better evaluate our method.
>
> We conducted pretraining for a 60M-parameter Llama model on 5 billion tokens and evaluated performance using validation perplexity. Our observations indicate a direct correlation among **the number of terms in the Neumann series**, the orthogonality approximation error and the resulting validation perplexity. The specific number of terms selected for our method represents an optimal trade-off between computational efficiency and model performance. The results are summarized in the following table. The corresponding orthogonality approximation error is calculated with the following formula: $e_{\text{orth}} = \frac{\| R{R}^T - {I} \|_F}{\| {I} \|_F}$.
>
> ***Ablation Study on Neumann Series Terms***
>
> | Scheme | Perplexity | Orth Error (max) |
> | :--- | :--- | :--- |
> | $k=0$ | Not converged | 0.127 |
> | $k=1$ | 22.56 | 0.026  |
> | $k=2$ | 21.54 | 0.021 |
> | $k=3$ | 20.22 | 0.017 |
> | $k=4$ | **20.19** | **0.013** |
>
> To answer the question of how the **block size** affects the validation perplexity, we first direct the reviewer’s attention to Tables 6 and 7 in the main text, which present the validation perplexity for 3B-parameter Llama models pretrained with block sizes of $256$ and $512$. These results are summarized in the table below:
>
> ***Ablation Study: Block Size (Llama 3B on 60B tokens)***
>
> | Method | Params (M) | Mem (G) | Val PPL |
> | :--- | :--- | :--- | :--- |
> | POET-X (b=256) | 366.64 | 60.58 | 12.76 |
> | POET-X (b=512) | 570.06 | 68.52 | **12.05** |
> | POET-XQ (b=256) | 366.64 | **51.66** | 16.21 |
> | POET-XQ (b=512) | 570.06 | 60.65 | 14.78 |
>
> To further address the reviewer’s question, we conducted an ablation study varying the block size during the pretraining of a 60M-parameter Llama model on 5 billion tokens. The results, summarized in the table below, demonstrate a clear correlation between block size and final validation perplexity. Notably, a block size of 256 performs comparably to the AdamW baseline while maintaining significantly higher computational and memory efficiency. For small block sizes to achieve better PPL, it requires more training iterations (ie, training tokens).
>
> ***Ablation Study: Block Size (Llama 60M on 5B tokens)***
>
> | Method | Val PPL |
> | :--- | :--- |
> | POET-X (b=64) | 35.81 |
> | POET-X (b=128) | 31.60  |
> | POET-X (b=256) | 28.71 |
> | POET-X (b=512) | 27.14 |
>
> Regarding the **recomputation overhead**, we point the reviewer to **Table 10** of the main paper. While the recomputation costs result in a marginal reduction in the throughput compared to POET-X (fast), this is significantly offset by its superior memory efficiency. As shown in **Table 9** of the main paper, POET-X (mem) requires substantially less memory than standard optimizers, including AdamW and Muon, as well as specialized memory-efficient optimizers such as GaLore/Q-GaLore and Apollo/Q-Apollo.
>
> ---
>
> **Q4: Headline claims such as “pretraining up to 13B on a single H100” would be more convincing if accompanied by a fuller specification of the training setup.**
>
> We agree that additional training details can strengthen our paper. For a single H100 to pretrain 13B LLM, the setting is identical to **Table 9** of the main paper. One can observe that it is more than sufficient for all POET-X variants to fit in a H100 GPU (for pretraining Llama-13B). The detailed settings are put in **Appendix A** of the main paper. We will supplement full details and release our training script for full reproducibility.

---

> > ### Author Rebuttal · Reviewer_kKuH · 2026-04-08
> >
> > Thank you for the rebuttal. I still find that the paper lacks sufficient theoretical analysis, and the contribution appears primarily engineering-oriented rather than algorithmically or theoretically substantive. Therefore, I keep my original rating of the paper.

---

> > > ### Author Response · Authors · 2026-04-08
> > >
> > > Thanks for the reply! We are deeply appreciative of the reviewer's time and efforts. To help us better understand and address the reviewer’s concerns, we would greatly appreciate it if the reviewer could kindly point us to the specific questions. We will try our best to address the reviewer's remaining concerns.
> > >
> > > With all due respect, we disagree with the characterization of our paper as primarily engineering-oriented. Our work achieves stable and efficient LLM pretraining through a spectrum-preserving algorithm and a joint algorithm-system co-design, where the engineering efforts, together with the underlying numerical and algebraic designs, are not ends in themselves but the means by which our method delivers strong memory efficiency and training stability.

---

### Official Review · Reviewer_2t2Q · 2026-03-14

**Soundness:** 4
**Presentation:** 3
**Significance:** 4
**Originality:** 3
**Overall Recommendation:** 5
**Confidence:** 3

**Summary:**

This work proposes a new variant of Reparameterized Orthogonal Equivalence Training (POET) that is just as good in terms of quality but more efficient in terms of resource (memory) usage.

**Compliance With Llm Reviewing Policy:**

Affirmed.

**Final Justification:**

Thanks to the authors for the response! I maintain the ratings.

**Key Questions For Authors:**

- I'm not entirely sure about the placement of the proposed method. Is it more like a new optimizer or a new efficient training algorithm? The experiments comparing it with both Muon and GaLore suggest they belong to different categories.
- In Tables 6 and 7, are the trainable parameters of the model or just of the optimizer (aka optimizer state) size?
- Table 6. Am I understanding correctly that the proposed model has lower perplexity while being four times smaller than Transformer with Adam?

**Limitations:**

Yes, for societal impact, but no explicit section on limitations.

**Strengths And Weaknesses:**

Strength:
- This work tackles an important issue of scalable and efficient LLM training.
- It applies sophisticated analysis to improve algorithms.
- The authors did a commendable job implementing custom CUDA and Triton kernels for notable speedups.
- The experiments are thorough, addressing both quality and efficiency to show the effectiveness of the proposed methods.
- Overall, I believe this is a sophisticated idea and very well executed by the authors.

Weakness:
- My main worry is that this proposed approach depends too much on one specific paper (POET), which is relatively new. It's not clear how the methods extend beyond this particular work.
- I find the paper quite dense and not the easiest to follow, but I wouldn't hold this against the author.

---

> ### Author Rebuttal · Authors · 2026-03-31
>
> We sincerely thank the reviewer for the constructive comments on our work. We take every comment seriously and hope our response can address the reviewer’s concerns.
>
> ---
>
> **Q1: this proposed approach depends too much on one specific paper (POET), which is relatively new. It's not clear how the methods extend beyond this particular work.**
>
> Thanks for raising this concern. While POET provides the conceptual and algorithmic foundation of this work, POET-X goes well beyond the original method. First, POET-X is among the first pretraining algorithms to achieve both superior memory efficiency and better perplexity than AdamW, which is a meaningful result in the current landscape of pretraining methods. Second, we do not see building on a relatively new method as a limitation, but as an opportunity. The POET framework remains underexplored and has substantial untapped potential, both theoretically and practically. POET-X shows that this line of research is not limited to the original POET algorithm, but can evolve into a scalable, efficient, and practically competitive training framework. Third, this paper also makes several system-level contributions that are independently valuable for scaling orthogonal transformation-based training.
>
> ---
>
> **Q2: I find the paper quite dense and not the easiest to follow, but I wouldn't hold this against the author.**
>
> We agree with the reviewer that the technical complexity of our method can be challenging to navigate within the current page constraints. Because our contributions lie in the combination of algorithms and efficient system realization, it requires a lot of technical details for the paper to be self-contained. We will utilize the additional camera-ready pages to enhance the manuscript’s clarity and flow. Furthermore, the POET-X implementation will be made publicly available.
>
> ---
>
> **Q3: I'm not entirely sure about the placement of the proposed method. Is it more like a new optimizer or a new efficient training algorithm?**
>
> Great question! We view POET-X primarily as a **reparameterized training algorithm**. It does function as an optimizer and performs multiplicative weight updates. However, unlike standard optimizers, POET-X updates the reparameterized components of the weights rather than the weight matrix itself. Furthermore, the stochastic nature of these components at each iteration enhances training expressivity, enabling more flexible and robust orthogonal transformations.
>
> Our baseline selection reflects POET-X’s dual focus on state-of-the-art memory efficiency while achieving high performance:
>
> - **Efficiency (GaLore, Apollo):** Benchmarks against current low-memory standards that often sacrifice performance.
>
> - **Performance (AdamW, Muon):** Demonstrates that POET-X matches the convergence and perplexity of industry-standard, full-memory optimizers despite its significantly reduced footprint.
>
> ---
>
> **Q4: In Tables 6&7, are the trainable parameters of the model or just of the optimizer state? Am I understanding correctly that the proposed model has lower perplexity while being 4 times smaller than Adam?**
>
> All evaluated methods, including POET-X, utilize the same 3B Llama base architecture; the "Params (M)" column refers specifically to the number of *actively updated* parameters during pretraining.
>
> - **AdamW/Muon/GaLore/Apollo:** These methods update the full weight matrices, requiring gradients and optimizer states for all ~2.7B parameters.
>
> - **POET-X:** This approach freezes the base weights and updates only the reparameterized orthogonal matrices. With a block size of 512, this results in only ~570M trainable parameters. Though the number of active parameters is small, expressiveness remains high because they are stochastically assigned per iteration. The key intuition is that any orthogonal matrix can be approximated by a product of sparse orthogonal matrices (e.g., Givens rotations); POET-X need only learn a sparse orthogonal matrix at each iteration (hence fewer active parameters yet high expressiveness).
>
> This reduction in active parameters is the primary mechanism that allows POET-X to achieve PEFT-like memory efficiency during full-scale pretraining. We have updated the table captions and included explicit memory footprint measurements in the revised manuscript to ensure clarity.
>
> ***Updated Table 6: PPL***
>
> | Method | Params (M) | Mem (G) | Val PPL |
> | :--- | :---: | :---: | :---: |
> | AdamW | 2764.47 | 81.03 | 12.69 |
> | Muon | 2764.47 | 70.94 | **11.45** |
> | APOLLO | 2764.47 | 80.60 | 12.97 |
> | GaLore | 2764.47 | 74.50 | 14.88 |
> | POET-X (b=256) | 366.64 | **60.58** | 12.76 |
> | POET-X (b=512) | 570.06 | 68.52 | 12.05 |
>
> ***Updated Table 7: Quantized PPL***
>
> | Method | Params (M) | Mem (G) | Val PPL |
> | :--- | :---: | :---: | :---: |
> | Quantized 8-bit APOLLO | 2764.47 | 66.37 | 20.49 |
> | Quantized 8-bit GaLore | 2764.47 | 66.28 | 17.74 |
> | POET-XQ (b=256) | 366.64 | **51.66** | 16.21 |
> | POET-XQ (b=512) | 570.06 | 60.65 | **14.78** |

---

> > ### Author Rebuttal · Reviewer_2t2Q · 2026-04-03
> >
> > Thanks to the authors for the response!

---

> > > ### Author Response · Authors · 2026-04-04
> > >
> > > We are very happy that we were able to address all of your concerns and want to sincerely thank the reviewer for their time, rigorous evaluation, and constructive feedback throughout this process.

---

### Decision · Program_Chairs · 2026-04-30

**Decision:**

Accept (spotlight)

**Comment:**

This paper presents POET-X, a scalable and memory-efficient variant of POET for large-scale LLM pretraining. The work addresses a highly important problem in modern ML systems: how to make full-scale pretraining substantially more memory efficient without sacrificing training quality or stability. The core contribution is a thoughtful algorithm–systems co-design that reformulates the original POET computation into a practical input-centric implementation and combines it with several nontrivial kernel- and parameterization-level optimizations. The resulting method significantly improves the practicality of spectrum-preserving training and pushes it to a scale that is clearly relevant to the community.

Overall, I view this as a strong paper with both substantive technical merit and clear practical significance. In particular, the empirical results are compelling: the paper reports major gains in memory efficiency and runtime relative to the original POET, enables pretraining of very large models under hardware constraints where standard baselines fail, and remains competitive in validation perplexity, in several settings outperforming AdamW while operating at a much lower memory footprint. The ability to pretrain LLMs up to the reported scale on a single H100 is an especially notable practical result. The quantized variant further strengthens the impact of the work.

The review outcome is strongly positive overall. The main remaining negative review argues that the contribution is primarily engineering-oriented and lacks sufficient theoretical novelty. I understand this perspective, but I do not find it sufficient to outweigh the paper’s strengths. For this submission, the primary scientific contribution is not a new learning theory result, but rather a principled and nontrivial reformulation that turns an otherwise impractical training paradigm into a scalable one. In the context of ICML, this is a meaningful contribution: the paper identifies concrete algorithmic bottlenecks, redesigns the computation in a mathematically grounded way, and demonstrates substantial gains at realistic scales. Moreover, the final paper is not limited to profiling or low-level optimization alone; it also presents a coherent training methodology with strong end-to-end evidence on large-model pretraining. I therefore view the “engineering” criticism as a matter of emphasis rather than a fatal weakness.

There are, of course, some limitations. The method currently relies on custom Triton/CUDA implementations, and broader portability across hardware backends will matter for long-term adoption. The paper would also benefit from a clearer discussion of limitations and from improved exposition in a few places. In addition, while POET-X is highly competitive and often superior to AdamW under memory constraints, Muon remains stronger in some perplexity comparisons, suggesting there is still room to improve the optimization side of the framework. However, these are appropriate limitations for a strong conference paper rather than reasons for rejection. Importantly, none of them undermine the central claims of the paper.

Taking everything together, I recommend Strong Accept. The paper tackles an important problem, introduces a technically substantive and well-executed solution, and provides strong empirical evidence at a scale that makes the contribution meaningful to the broader ICML audience. I also believe it is worthy of spotlight / oral consideration, as efficient and scalable LLM pretraining is a central topic, and this paper combines conceptual clarity, systems relevance, and practical impact in a way that should attract broad interest.